# RANKING IS REWARD: INTRA-GROUP PREFERENCE RANKING FOR GROUP RELATIVE POLICY OPTIMIZATION

## ABSTRACT

Recent breakthroughs in Large Language Model (LLM) reasoning have been driven by reinforcement learning techniques like PPO and GRPO. However, in Reinforcement Learning with Verifiable Rewards (RLVR), sparse rewards hinder learning when group samples receive identical scores. While existing methods attempt to address this with data filtering, they inadvertently limit progress on correctly answered prompts. Additionally, reward models based on absolute numerical scores often suffer from range instability, undermining training stability. To address these issues, we introduce intra-group response preference ranking as a reward signal. We propose the Ranking Reward Model (RRM), a listwise preference model designed for GRPO, which outputs relative preference rankings for multiple responses to a single prompt. RankGRPO incorporates three strategies to leverage these rankings, mitigating vanishing gradients and instability from absolute scoring. Experiments show that RankGRPO improves performance across RLVR benchmarks, open-ended tasks, and reward model evaluations. RRM, trained with limited data, outperforms traditional numerical reward models trained on larger datasets, demonstrating the potential of RankGRPO and the effectiveness of ranking-based reward signals. Our source code is available at https://anonymous.4open.science/r/RankGRPO-0542.

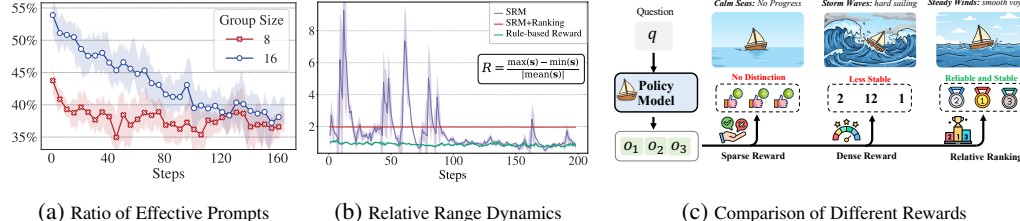

| (a) Ratio of Effective Prompts | (b) Relative Range Dynamics | (c) Comparison of Different Rewards |

Figure 1: Comparative analysis of reward formulations. (a) Sparse rewards cause zero intra-group variance, reducing data utilization efficiency during GRPO training. (b) Scalar reward models exhibit higher numerical dispersion, compromising stability in intra-group advantage estimation. (c) RankGRPO utilizes relative ranking to sustain stable advantages through consistent gradient signals.

## 1 INTRODUCTION

In recent years, reinforcement learning (RL) has driven significant advances in natural language processing (NLP), reshaping the reasoning paradigms of large language models (LLMs). Through large-scale RL training, models such as DeepSeek-R1 (Guo et al., 2025) and OpenAI O1 (Jaech et al., 2024) have demonstrated sophisticated reasoning abilities including self-verification and iterative refinement, which substantially improve their performance on challenging mathematical and programming tasks. Building on this progress, Group Relative Policy Optimization (GRPO) (Shao et al., 2024) has emerged as a key method for scaling LLMs during testing. By introducing an intra-group relative evaluation mechanism, GRPO reduces the bias of value function estimation and alleviates the heavy memory requirements associated with traditional Proximal Policy Optimization

(PPO) (Schulman et al., 2017), providing a more efficient and robust training paradigm for the next generation of LLMs.

Reward modeling has become a central component in RL, providing reliable signals that guide LLM responses (Gao et al., 2023). High-quality and robust rewards significantly enhance performance in specialized domains (Lightman et al., 2023b; Liu et al., 2025c). Yet modeling intermediate rewards within Chain-of-Thought (CoT) reasoning remains inherently difficult (Chen et al., 2025), which leaves most test-time scaling approaches dependent on rule-based functions that deliver sparse feedback only on final correctness (Yu et al., 2025). However, as GRPO training progresses, an increasing fraction of sample groups reach unanimous correctness, thereby reducing the proportion of effective updates. In this work, we call a training prompt an *effective sample* if its group of $G$ sampled responses contains both correct and incorrect outputs under the rule-based verifier. Only such prompts induce non-zero within-group reward variance and thus contribute non-trivial policy gradients, whereas prompts whose $G$ responses are all correct or all incorrect are treated as ineffective. As shown in Figure 1a, which plots the fraction of effective samples along the GRPO training trajectory, fewer than **40%** of samples provide meaningful gradients in later stages, leading to wasted computation and constraining the model's ability to acquire stronger reasoning strategies. This phenomenon is most salient when the problem difficulty is moderate, yet the absolute fraction of effective samples remains relatively low across different types of prompts.

Reward models help reduce sparse rewards but still have structural limitations. Outcome reward models (ORMs) judge only the final answer and overlook reasoning quality, while process reward models (PRMs) guide intermediate steps but require costly annotation. Most of these models are scalar reward models (SRMs). As illustrated in Figure 1b, the relative range of SRM scores exhibits a significantly higher magnitude compared to ranking-based methods. This indicates a high degree of numerical dispersion which undermines the stability of intra-group advantage estimation. Since GRPO relies on within-group comparisons, the critical factor is the relative rank. Although absolute scoring is capable of distinguishing responses, it introduces unnecessary numerical variance. Consequently, this approach leads to unstable gradients due to its sensitivity to score magnitude and task difficulty.

In this work, we propose RankGRPO, a novel algorithm that enhances GRPO by integrating ranking information among intra-group samples during advantage computation. This approach is designed to overcome the dual challenges of sparse rewards failing to provide sustained, valuable gradients and the instability of traditional numerical reward models. Specifically, we devise three distinct mechanisms that incorporate relative ranking and rule-based rewards to varying degrees: Ranking as Weight, Ranking as Supplement, and Ranking as Reward. To facilitate this ranking-based approach, we introduce the Ranking Reward Model (RRM). The RRM is trained to rank the quality of multiple input samples, a design motivated by GRPO's inherent mechanism of comparing responses within a group. We then employ the RRM within the RankGRPO framework to order the quality of these intra-group responses. This method enables RankGRPO to continue learning from data conventionally defined as invalid in standard GRPO, thereby significantly improving the model's reasoning quality while maximizing the utilization of rollout data. Through extensive experiments across diverse settings, we rigorously validate the effectiveness of our proposed methodology, demonstrating significant improvements over existing approaches. Our contributions are summarized as follows:

- We propose RankGRPO and design three distinct mechanisms to integrate intra-group relative ranking into advantage computation, addressing the limitations of sparse rewards.

- We propose the RRM, which performs relative ranking over multiple samples. This approach mitigates the instability challenges of numerical SRMs and is better suited to GRPO's framework for computing group-relative advantages.

- Extensive experiments on multiple tasks demonstrate the effectiveness of RankGRPO in enhancing data utilization efficiency and improving model performance.

## 2 RELATED WORK

**Inference-Time Scaling for LLMs.** Inference-time scaling complements training efforts, with research focusing on sampling and reward model aggregation (Brown et al., 2024; Snell et al., 2025; Wu et al., 2024). A key approach is Reinforcement Learning with Verifiable Reward (RLVR),

which improves reasoning by using external verifiers for reward signals instead of model-generated scores (Zeng et al., 2025). Methods like PPO (Schulman et al., 2017) and GRPO (Shao et al., 2024) are commonly used for policy optimization, driving further RL advancements in reasoning tasks (Kazemnejad et al., 2024; Yuan et al., 2025). Notable innovations include DAPO, which filters zero-variance prompts (Yu et al., 2025), and GRESO, which uses probabilistic pre-filtering (Zheng et al., 2025). While both improve data efficiency, DAPO incurs computational overhead, and GRESO may discard useful learning opportunities due to its simplistic reward structure.

**Reward Models.** Reward models (RMs) are pivotal in RL, especially for aligning large language models (LLMs) and scaling inference. Designed to capture human preferences, RMs complement rule-based rewards (Christiano et al., 2017; Ouyang et al., 2022). Mainstream RMs typically function as discriminative classifiers, providing scalar rewards to rank responses (Cai et al., 2024; Liu et al., 2025a; Lou et al., 2024). Other methods harness LLMs as judges, offering preference scores or critiques on generated content (Zheng et al., 2023). Approaches like Direct Preference Optimization (DPO) eliminate the need for explicit RMs, instead directly optimizing policies from preference pairs (Rafailov et al., 2023). Despite their advantages, RMs face challenges, such as the high cost of preference data, biases, and the risk of reward hacking (Gao et al., 2023; Skalse et al., 2022).

## 3 Preliminaries

To optimize the LLM policy, GRPO (Shao et al., 2024) introduces an alternative RL algorithm, which is a memory-efficient variant of PPO (Schulman et al., 2017). A notable feature of GRPO is that it typically operates without a learned value function. Instead, for a given prompt $p$, the current policy generates a group of $G$ responses $\{o_1, \ldots, o_G\}$. The rewards $\{s_1, \ldots, s_G\}$ for these responses are then used to compute the relative advantage for each response:

$$\hat{A}_k = \frac{s_k - \text{mean}(\{s_k | k = 1, 2, \ldots, G\})}{F_{\text{norm}}}. \tag{1}$$

Here, $F_{\text{norm}}$ serves as an optional normalization factor. In the standard GRPO implementation, $F_{\text{norm}}$ is defined as $\text{std}(\{s_k | k = 1, \ldots, G\})$. In contrast, alternative implementations in RLVR fix the normalization factor to unity so that $F_{\text{norm}} = 1$ (Liu et al., 2025b; Chu et al., 2025).

GRPO then maximizes a clipped surrogate objective function to ensure stable updates. Let $\pi_{\theta_{\text{old}}}$ represent the policy before the update. For each token $o_{k,t}$ in a trajectory $o_k$ (from state $s_t$), the importance sampling ratio is defined as $\rho_{k,t}(\theta) = \frac{\pi_\theta(o_{k,t}|s_t)}{\pi_{\theta_{\text{old}}}(o_{k,t}|s_t)}$. The objective is then given by:

$$\mathcal{J}_{\text{GRPO}}(\theta) = \frac{1}{G} \sum_{k=1}^{G} \frac{1}{|o_k|} \sum_{t=1}^{|o_k|} \left( \min \left( \rho_{k,t}(\theta) \cdot \hat{A}_k, \text{clip}(\rho_{k,t}(\theta), 1 - \epsilon, 1 + \epsilon) \cdot \hat{A}_k \right) \right), \tag{2}$$

where $\epsilon$ is a small hyperparameter defining the clipping range. This mechanism ensures that the LLM policy is updated while maintaining stable gradient constraints.

## 4 Methodology

To resolve the mismatch between absolute reward scores, whether rule-based or model-based, and the relative group-wise signal required by GRPO, we propose RankGRPO, a reinforcement learning framework built on intra-group preference rankings. We further introduce the RRM, tailored to GRPO, which mitigates the scale instability of SRMs by producing consistent relative orderings of responses within each group.

### 4.1 RankGRPO: Learning with Intra-Group Preference Rankings

We introduce RankGRPO, a framework that integrates intra-group relative quality rankings produced by a dedicated ranking model into GRPO to mitigate gradient vanishing when reward variance collapses within the response set $\{o_i\}_{i=1}^{G}$. Given the group of responses $\{o_i\}_{i=1}^{G}$, this ranking model assigns each response a rank $r_i$ that reflects its relative quality. For each response $o_i$ with rank $r_i$, we map the binary rule-based reward $s_i^{\text{rule}} \in \{0, 1\}$ to a rank-enhanced score $s_i^{\text{rank}} = f(s_i^{\text{rule}}, r_i)$. We

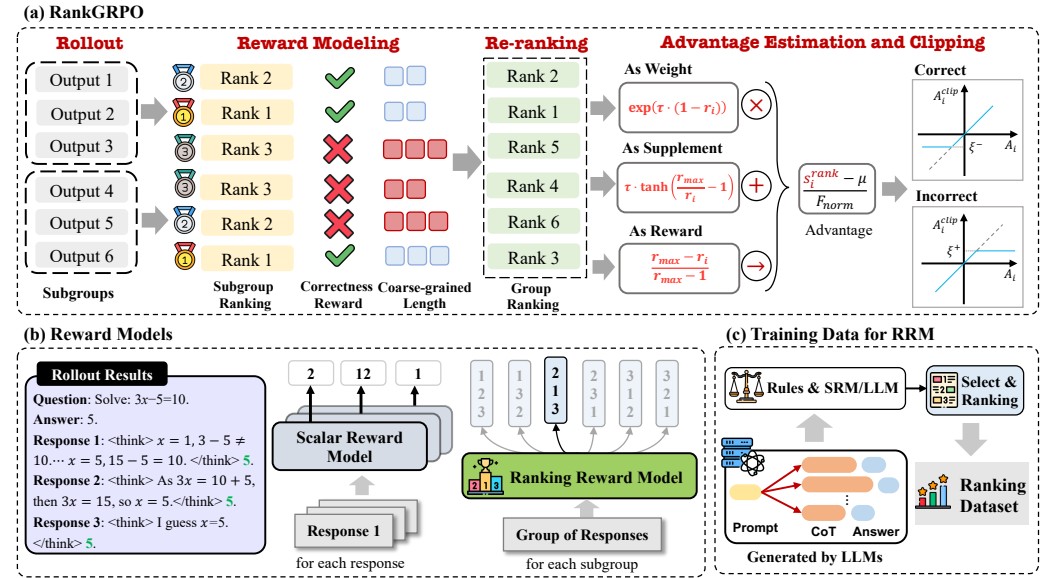

Figure 2: Overview of RankGRPO. (a) Workflow of RankGRPO, illustrating three strategies for integrating intra-group preference rankings with scalar reward scores during advantage estimation. (b) Reward-model comparison: the SRM assigns independent scores to each response, whereas the RRM produces a relative ordering of multiple responses within a group. (c) Training-data processing pipeline for the RRM.

instantiate $f(\cdot)$ with three strategies that leverage relative orderings to different degrees. These rank-aware adjustments counteract the failure mode of zero within-group reward variance and preserve informative gradients for optimization.

**Ranking as Weight.** We rescale the original reward scores according to the relative quality of responses within each group:

$$s_i^{\text{rank}} = f(s_i^{\text{rule}}, r_i) = \exp\left(\tau \cdot (1 - r_i)\right) \cdot s_i^{\text{rule}}, \tag{3}$$

where the coefficient $\tau$ controls the strength of rank-based weighting over ranks $r_i$, amplifying top-ranked samples and suppressing lower-ranked ones. This mechanism further differentiates correct samples, while incorrect samples are handled as in GRPO, enabling the model to continue learning from groups in which all responses are correct.

**Ranking as Supplement.** To leverage the signal from all samples, we add a bounded, rank-based correction to the rule reward. For item $i$ with within-group rank $r_i \in \{1, \ldots, r_{\max}\}$, we define

$$s_i^{\text{rank}} = f(s_i^{\text{rule}}, r_i) = s_i^{\text{rule}} + \tau \cdot \tanh\left(\frac{r_{\max}}{r_i} - 1\right), \tag{4}$$

where $\tau > 0$ controls the magnitude of the adjustment. It keeps the rule reward as the primary signal and introduces fine-grained distinctions among responses with identical rule correctness, ensuring that every sample contributes information.

**Ranking as Reward.** We replace the rule-based score with a normalized, rank-only reward that maps within-group ranks directly to $[0, 1]$. For item $i$ with rank $r_i \in \{1, \ldots, r_{\max}\}$, define

$$s_i^{\text{rank}} = f(s_i^{\text{rule}}, r_i) = \frac{r_{\max} - r_i}{r_{\max} - 1}. \tag{5}$$

This monotone mapping assigns a fixed score to each rank and is invariant to any affine transformation of the rule-base scores. Consequently, rewards depend solely on the relative ordering: higher-ranked responses always receive higher rewards, and the fixed per-rank values yield stable advantages across groups.

**Advantage of RankGRPO.** Directly using the mean of $s^{\text{rank}}$ as a baseline may lead to inconsistencies between the correctness of certain responses and the direction of their computed advantages.

To mitigate the potential negative impact of such misaligned advantages, we introduce a clipping strategy. Specifically, advantages that contradict correctness are truncated, thereby improving the stability of the training process. The clipping is defined as follows:

$$A_i^{\text{clip}} = \begin{cases} \max\big(A_i, \xi^-\big), & \text{if correct} \\ \min\big(A_i, \xi^+\big), & \text{if incorrect} \end{cases} \tag{6}$$

where $A_i$ denotes the advantage of the $i$-th sample, and $\xi^+/\xi^-$ are clipping thresholds that prevent excessively large positive or negative values. This design ensures that positive advantages are lower-bounded and negative ones are upper-bounded, aligning the reward signal with correctness and enhancing training robustness. For convenience, we denote the three mechanisms of RankGRPO as W, S, and R, representing *Ranking as Weight*, *Ranking as Supplement*, and *Ranking as Reward*, respectively, in the following experiments.

### 4.2 RANKING REWARD MODEL

In GRPO, advantage estimation depends on the relative relationships among responses within a group, computed through mean-normalized rewards. However, conventional reward models output scalar scores whose absolute values are difficult to control, even if they preserve the correct preference order. This mismatch makes the GRPO advantage unstable, since inconsistent score scales lead to unreliable within-group normalization. To address this issue, we introduce the Ranking Reward Model (RRM), which directly predicts the relative ordering of $n$ responses rather than assigning unconstrained scalar scores. This ranking-based formulation aligns naturally with the group-wise normalization in GRPO and eliminates the need for stable score magnitudes.

RRM is a sequence classification model fine-tuned from an LLM. We replace the language-model head with a classification layer whose classes correspond to the $n!$ possible permutations over the $n$ input responses, and we train the model using a standard cross-entropy loss. As illustrated in Figure 2, RRM outputs a relative ranking for $n$ responses. To construct training data, we first sample multiple CoT responses for each prompt. For tasks with verifiable correctness, a rule-based verifier assigns a binary correctness label to each response. We rigorously enforce that correct responses must be ranked higher than incorrect ones. Within each correctness category, we sort responses based on scores from an SRM to obtain a fine-grained order. The LLM judge is utilized only in rare instances where the SRM ranking contradicts the correctness label. In such cases, we discard the inconsistent SRM scores and query the LLM judge to provide a local ranking that respects the correctness constraint. For open-ended tasks without a reliable rule-based verifier, we directly sort responses by SRM scores without invoking an LLM judge. Training instances are then created by selecting $n$ responses from the finalized total order, randomly shuffling their input sequence, and using the resulting permutation index as the supervision label to mitigate positional bias.

### 4.3 HIERARCHICAL RE-RANKING

Given that RMs are more susceptible to reward hacking than rule-based rewards, we introduce a re-ranking phase after obtaining relative sample ordering via the RRM. This phase first sorts samples based on verifiable reward scores, then mitigates model overthinking by prioritizing shorter responses through length-based adjustment, and finally integrates the quality ordering from the ranking reward model. Since minor length variations among responses with identical reward scores do not capture quality distinctions, we apply coarse-grained length discretization for calibration, as formalized in Equation 7:

$$\mathcal{B}_i = \begin{cases} \left\lfloor \dfrac{\ell_i}{\lambda} \right\rfloor, & \text{if } o_i \text{ is correct} \\ +\infty, & \text{if } o_i \text{ is incorrect} \end{cases} \tag{7}$$

where $\ell_i$ is the length of response $o_i$, and hyperparameter $\lambda$ controls discretization granularity. When reward scores are equal, correct responses are reordered by ascending $\mathcal{B}_i$. Incorrect responses are assigned $\mathcal{B}_i = +\infty$, removing length constraints from their ranking.

Since RRM is applied to $n$ responses at a time, we choose the GRPO group size $G$ to be a multiple of $n$ and partition each group into $G/n$ disjoint subgroups, each containing exactly $n$ responses. As depicted in Figure 2, we apply RRM to every subgroup to obtain local relative ranks $r_i^{\text{rrm}}$ and

---

**Algorithm 1** RankGRPO

---

**Input** policy $\pi_\theta$, dataset $\mathcal{D}$, rule-based verifier $R_\phi$, ranking reward model $R_\psi$, group size $G$, subgroup size $n$, inner steps $\mu$.

1: **for** step = 1, ..., M **do**
2:     Sample a batch $\mathcal{D}_b$ from $\mathcal{D}$ and set $\pi_{\theta_{\text{old}}} \leftarrow \pi_\theta$
3:     **for** each $q \in \mathcal{D}_b$ **do**
4:         **Sampling:** Generate $G$ responses $o_i$ using $\pi_{\theta_{\text{old}}}$.
5:         **Reward calculation:** Compute rule-based rewards $s_i^{\text{rule}}$ and ranking-based relative ranks $r_i$ for each response $o_i$ using $R_\phi$ and $R_\psi$.
6:         **Hierarchical Re-ranking:** Sort responses lexicographically using the priority tuple $(s_i^{\text{rule}}, \mathcal{B}_i, r_i^{\text{rrm}})$ (Correctness $\rightarrow$ Length $\rightarrow$ RRM) to determine the global rank $r_i$, and compute final rewards $s_i^{\text{rank}}$ using the rank-mapping function $f(s_i^{\text{rule}}, r_i)$.
7:         **Advantage computation and clipping:** Compute group-relative advantages (Equation (1)) and clip the advantages based on a threshold $\xi$ (Equation (6)).
8:         **Policy update:** Update the policy $\pi_\theta$ by maximizing the GRPO objective (Equation (2)).
9:     **end for**
10: **end for**
11: **Output:** The final policy $\pi_\theta$

---

then perform a hierarchical re-ranking stage to produce the final ranks $r_i$ for all $G$ responses. The complete RankGRPO workflow is formalized in Algorithm 1. This approach integrates intra-group relative ordering into GRPO, thereby mitigating gradient vanishing induced by sparse rewards while circumventing reliability limitations of conventional reward models and enabling priority aware multi objective optimization by merging correctness, reasoning efficiency, and chain of thought quality into a single unified ranking.

## 5 EXPERIMENTS

To comprehensively evaluate the performance of RankGRPO and the effectiveness of our proposed RRM, we conduct experiments from three perspectives:

- **Tasks with Verifiable Rewards**: We select mathematical reasoning and logical reasoning tasks to assess the model's capabilities in scenarios where rewards can be explicitly verified.

- **Open-ended Writing Tasks**: We utilize WritingBench (Wu et al., 2025) to evaluate the model's performance on open-domain writing challenges, which covers 6 core domains and 100 subdomains, encompassing a diverse range of writing tasks and styles.

- **Evaluation of Reward Models**: We evaluate reward models with a Reward-guided Test-Time Scaling framework (Zou et al., 2025), where each model selects the most accurate solution from candidates, and accuracy of the chosen solutions serves as the metric.

### 5.1 TASKS WITH VERIFIABLE REWARDS

**Baselines.** We conduct our experiments on DeepSeek-R1-Distill-Qwen-1.5B and DeepSeek-R1-Distill-LlaMA-8B (Guo et al., 2025). Our primary comparison is against four recent state-of-the-art reinforcement learning methods: (1) GRPO (Shao et al., 2024), (2) Dr.GRPO (Liu et al., 2025b), (3) GPG (Chu et al., 2025), and (4) DAPO (Yu et al., 2025). Our RRM is trained on 25k data using Qwen2.5-7B-Instruct-1M (Team, 2025).

**Datasets.** For RL training, we use approximately 16,000 mathematics and logic samples filtered by difficulty from the GURU dataset (Cheng et al., 2025), and for the 8B model, we additionally include the Open-RS dataset (Dang & Ngo, 2025). For ablation and analysis experiments, we train the 1.5B model using the SimpleRL dataset (Zeng et al., 2025). For evaluation, we employ several challenging mathematical and logical reasoning benchmarks to assess our models' performance. Detailed descriptions and references for all evaluation datasets are provided in Appendix B.3.

**Performance.** Table 1 shows that RankGRPO achieves competitive or superior results on mathematical reasoning. On logic reasoning tasks, it consistently outperforms all baselines, underscoring the benefit of ranking-based optimization. Across variants, *Ranking as Weight* performs slightly

Table 1: Overall performance on eight competition-level mathematical reasoning benchmarks and two logic reasoning benchmarks. We report the mean score $\pm$ confidence interval. The average response length (tokens) is reported in the rightmost column. **Bold** and underlined indicate the best and second-best performance, respectively.

| | Method | AIME 24 $Avg@32$ | AIME 25 $Avg@32$ | MATH 500 $Avg@4$ | GSM8K $Avg@4$ | Olympiad $Avg@4$ | GaoKao $Avg@4$ | Minerva $Avg@4$ | AMC $Avg@16$ | Avg | Zebra $Avg@4$ | Ordering $Avg@4$ | Avg | Avg Len. (Tokens) |
|---|---|---|---|---|---|---|---|---|---|---|---|---|---|---|
| R1-Distill-Qwen-1.5B | Baseline | $28.5_{\pm1.0}$ | $23.5_{\pm2.2}$ | $82.7_{\pm2.1}$ | $85.8_{\pm1.1}$ | $43.3_{\pm1.3}$ | $71.7_{\pm1.4}$ | $26.5_{\pm0.7}$ | $63.1_{\pm0.5}$ | $53.1_{\pm0.7}$ | $0.7_{\pm0.2}$ | $14.0_{\pm1.2}$ | $7.3_{\pm0.7}$ | $11727_{\pm205}$ |
| | GRPO | $30.8_{\pm1.1}$ | $23.7_{\pm0.6}$ | $83.5_{\pm0.9}$ | $86.3_{\pm1.0}$ | $44.5_{\pm0.3}$ | $\mathbf{74.5}_{\pm0.3}$ | $28.0_{\pm3.3}$ | $65.9_{\pm1.2}$ | $54.6_{\pm0.4}$ | $2.5_{\pm0.4}$ | $22.4_{\pm1.6}$ | $12.4_{\pm1.0}$ | $7253_{\pm166}$ |
| | Dr.GRPO | $29.7_{\pm0.6}$ | $23.6_{\pm1.9}$ | $83.7_{\pm1.4}$ | $\mathbf{86.9}_{\pm2.8}$ | $45.4_{\pm1.7}$ | $73.6_{\pm0.7}$ | $27.4_{\pm1.5}$ | $65.9_{\pm2.0}$ | $54.5_{\pm0.4}$ | $3.0_{\pm0.9}$ | $20.4_{\pm3.2}$ | $11.1_{\pm4.2}$ | $7239_{\pm149}$ |
| | GPG | $\underline{32.0}_{\pm0.7}$ | $\mathbf{24.6}_{\pm0.3}$ | $84.7_{\pm0.9}$ | $\underline{86.7}_{\pm0.3}$ | $45.5_{\pm1.2}$ | $73.6_{\pm1.1}$ | $28.1_{\pm0.4}$ | $65.2_{\pm1.7}$ | $55.1_{\pm0.7}$ | $3.8_{\pm1.3}$ | $22.2_{\pm4.6}$ | $13.5_{\pm3.9}$ | $9937_{\pm173}$ |
| | DAPO | $30.1_{\pm3.5}$ | $22.8_{\pm0.1}$ | $\underline{84.3}_{\pm0.5}$ | $86.4_{\pm0.2}$ | $45.5_{\pm1.6}$ | $\underline{74.2}_{\pm0.2}$ | $\underline{29.2}_{\pm1.5}$ | $\underline{67.7}_{\pm1.6}$ | $55.0_{\pm0.2}$ | $6.8_{\pm2.7}$ | $28.4_{\pm3.4}$ | $17.6_{\pm3.0}$ | $10006_{\pm197}$ |
| | RankGRPO(W) | $30.7_{\pm0.7}$ | $23.7_{\pm1.2}$ | $83.1_{\pm1.7}$ | $85.8_{\pm0.5}$ | $44.8_{\pm1.0}$ | $73.8_{\pm0.8}$ | $\mathbf{29.7}_{\pm0.2}$ | $67.4_{\pm1.5}$ | $54.9_{\pm1.0}$ | $\underline{9.8}_{\pm2.9}$ | $\underline{34.9}_{\pm2.2}$ | $\underline{22.3}_{\pm2.5}$ | $\mathbf{6226}_{\pm137}$ |
| | RankGRPO(S) | $31.2_{\pm0.3}$ | $\underline{24.1}_{\pm0.6}$ | $\mathbf{84.8}_{\pm0.4}$ | $86.3_{\pm0.3}$ | $\mathbf{46.4}_{\pm0.8}$ | $74.1_{\pm0.2}$ | $28.6_{\pm0.3}$ | $66.6_{\pm0.7}$ | $\mathbf{55.3}_{\pm0.4}$ | $\mathbf{10.1}_{\pm0.9}$ | $\mathbf{39.5}_{\pm6.9}$ | $\mathbf{24.6}_{\pm3.5}$ | $\underline{6601}_{\pm159}$ |
| | RankGRPO(R) | $\mathbf{32.3}_{\pm0.7}$ | $22.7_{\pm1.3}$ | $83.4_{\pm0.8}$ | $86.6_{\pm0.9}$ | $\underline{46.4}_{\pm2.7}$ | $74.1_{\pm1.1}$ | $28.1_{\pm0.5}$ | $\mathbf{68.1}_{\pm0.6}$ | $\underline{55.2}_{\pm0.7}$ | $6.1_{\pm2.9}$ | $31.4_{\pm2.6}$ | $18.8_{\pm1.9}$ | $7121_{\pm190}$ |
| R1-Distill-LLaMA-8B | Baseline | $47.8_{\pm1.6}$ | $32.3_{\pm0.5}$ | $90.0_{\pm0.9}$ | $90.0_{\pm0.9}$ | $51.5_{\pm2.2}$ | $77.0_{\pm1.0}$ | $29.6_{\pm2.7}$ | $78.5_{\pm2.4}$ | $62.1_{\pm1.2}$ | $9.0_{\pm3.7}$ | $55.8_{\pm1.5}$ | $32.4_{\pm1.5}$ | $9209_{\pm419}$ |
| | GRPO | $50.1_{\pm1.7}$ | $33.2_{\pm1.2}$ | $90.2_{\pm1.6}$ | $91.7_{\pm0.7}$ | $56.6_{\pm1.9}$ | $80.3_{\pm0.8}$ | $30.4_{\pm1.5}$ | $81.7_{\pm1.5}$ | $64.3_{\pm0.4}$ | $30.8_{\pm1.2}$ | $77.7_{\pm1.4}$ | $54.3_{\pm1.3}$ | $6638_{\pm157}$ |
| | Dr.GRPO | $50.9_{\pm1.3}$ | $32.7_{\pm0.5}$ | $89.8_{\pm2.6}$ | $90.3_{\pm2.5}$ | $54.1_{\pm1.3}$ | $79.4_{\pm1.3}$ | $30.2_{\pm1.3}$ | $\underline{83.1}_{\pm1.0}$ | $63.8_{\pm1.2}$ | $23.5_{\pm1.3}$ | $74.5_{\pm1.5}$ | $49.0_{\pm0.4}$ | $6667_{\pm375}$ |
| | GPG | $51.0_{\pm1.3}$ | $32.4_{\pm2.3}$ | $89.1_{\pm1.2}$ | $91.3_{\pm1.4}$ | $54.7_{\pm1.7}$ | $80.1_{\pm0.7}$ | $30.0_{\pm1.7}$ | $82.1_{\pm2.1}$ | $63.8_{\pm0.7}$ | $24.6_{\pm3.0}$ | $78.8_{\pm1.0}$ | $51.7_{\pm1.6}$ | $7228_{\pm352}$ |
| | DAPO | $47.8_{\pm2.3}$ | $34.6_{\pm1.6}$ | $90.0_{\pm1.0}$ | $\underline{92.5}_{\pm1.3}$ | $58.1_{\pm0.6}$ | $\underline{81.7}_{\pm0.9}$ | $32.9_{\pm1.7}$ | $79.4_{\pm1.9}$ | $64.6_{\pm0.9}$ | $37.3_{\pm1.5}$ | $\underline{89.3}_{\pm0.9}$ | $63.3_{\pm1.2}$ | $6495_{\pm309}$ |
| | RankGRPO(W) | $\underline{51.1}_{\pm1.5}$ | $\underline{35.1}_{\pm1.0}$ | $89.4_{\pm2.5}$ | $91.7_{\pm2.2}$ | $\underline{59.1}_{\pm1.4}$ | $81.1_{\pm0.9}$ | $31.2_{\pm1.1}$ | $82.6_{\pm1.8}$ | $\underline{65.2}_{\pm0.6}$ | $\mathbf{40.7}_{\pm2.9}$ | $\mathbf{88.9}_{\pm2.1}$ | $\mathbf{64.8}_{\pm0.5}$ | $5530_{\pm167}$ |
| | RankGRPO(S) | $50.9_{\pm1.1}$ | $\mathbf{35.4}_{\pm1.7}$ | $\mathbf{91.5}_{\pm1.3}$ | $\mathbf{92.5}_{\pm1.2}$ | $\mathbf{59.5}_{\pm0.9}$ | $\mathbf{82.5}_{\pm1.2}$ | $\mathbf{33.1}_{\pm1.2}$ | $\mathbf{83.6}_{\pm0.5}$ | $\mathbf{66.0}_{\pm1.2}$ | $\underline{40.4}_{\pm3.4}$ | $89.2_{\pm0.5}$ | $\underline{64.7}_{\pm1.4}$ | $\mathbf{5370}_{\pm201}$ |
| | RankGRPO(R) | $\mathbf{51.9}_{\pm1.4}$ | $34.8_{\pm1.1}$ | $\underline{90.5}_{\pm1.4}$ | $91.7_{\pm0.7}$ | $55.3_{\pm1.3}$ | $81.7_{\pm1.1}$ | $\underline{32.9}_{\pm1.0}$ | $81.6_{\pm1.5}$ | $65.1_{\pm0.4}$ | $35.8_{\pm1.0}$ | $88.0_{\pm0.9}$ | $61.9_{\pm0.9}$ | $\underline{5479}_{\pm315}$ |

worse as it ignores groups with all incorrect responses, whereas others leverage both all-correct and all-incorrect groups for stronger training signals. In particular, *Ranking as Supplement*, which emphasizes rule-based rewards in advantage estimation, yields more stable improvements on tasks with verifiable correctness. Overall, these results demonstrate the effectiveness of RankGRPO in enhancing both mathematical and logical reasoning performance across a range of tasks.

**In-depth Comparison on Mathematical Reasoning.** Figure 3a presents a comparison between our models and several open-source reasoning baselines. Our approach demonstrates a superior balance between efficiency and performance. Notably, it surpasses resource-intensive baselines (e.g., STILL-3-1.5B) with significantly reduced data and compute costs, while also outperforming other open-source models in accuracy. This demonstrates that incorporating *intra-group relative ranking* effectively compensates for the limitations of rule-based rewards in GRPO, thereby improving the achievable upper bound of the algorithm.

We further investigate different strategies when using only reward models. As shown in Figure 3d, numerical reward models suffer from instability, occasionally degrading performance on certain datasets. By contrast, converting numerical scores into relative rankings significantly stabilizes the reward signal and leads to consistent performance gains. Figure 3b and 3c illustrate this effect from the perspective of training dynamics, highlighting how relative rewards mitigate fluctuations. Moreover, when adopting our proposed *RRM*, the performance is further improved, providing additional evidence that ranking-based rewards offer a more reliable and effective training signal than absolute numerical values. This demonstrates that relative ranking-based rewards enhance both stability and performance across various tasks.

## 5.2 Open-ended Writing Tasks

**Datasets and Evaluation.** To train our RRM, we first construct a preference dataset. Following the methodology described in Figure 2c, we select 10,000 samples from the Dolphin-R1 dataset[1] and generate ranked responses using models of varying scales. For model fine-tuning, we utilize a separate set of 22,000 samples to fine-tune the Qwen3-1.7B model (Yang et al., 2025).

**Baselines.** We adopt GRPO (Shao et al., 2024) as our primary algorithmic baseline for comparison. In addition, we benchmark our RRM against several state-of-the-art reward models: (1) Skywork-Reward-V2-Llama-3.1-8B (Liu et al., 2025a), (2) URM-LLaMa-3.1-8B (Lou et al., 2024).

---

[1] https://huggingface.co/datasets/QuixiAI/dolphin-r1

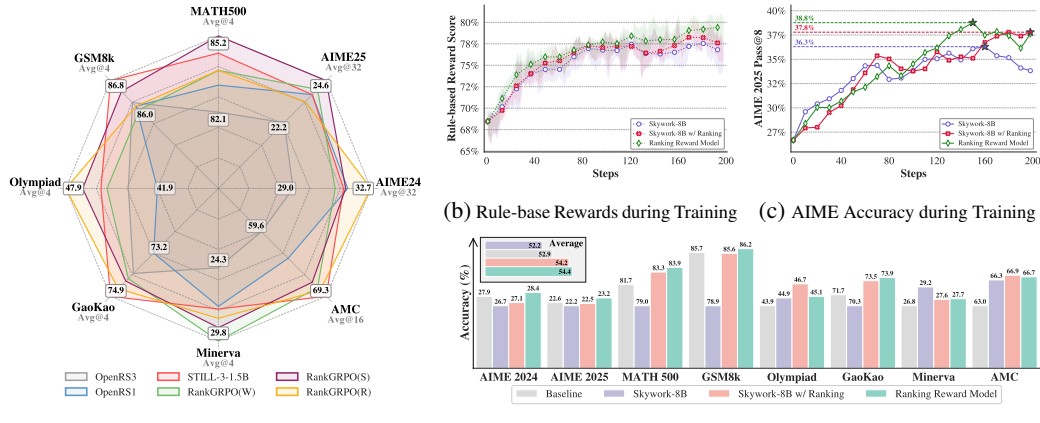

(a) Comparison with Open-Source Reasoning Models

(b) Rule-base Rewards during Training

(c) AIME Accuracy during Training

(d) Performance on Mathematical Reasoning with Different Reward Models

Figure 3: A deeper comparison of RankGRPO. **Left** (a): Results against existing models. **Right** (b-d): Training dynamics and reward analyses.

Table 2: Overall performance on Writing Bench. We report the mean score and confidence intervals ($\pm$). The '$\hookrightarrow$' symbol denotes a variant of the method listed directly above. **Bold** and underline mark the best and second-best results, respectively.

| METHOD | OVERALL | ACADEMIC & ENGINEERING | FINANCE & BUSINESS | POLITICS & LAW | LITERATURE & ARTS | EDUCATION | ADVERTISING & MARKETING |
|---|---|---|---|---|---|---|---|
| Qwen3-1.7B | $70.06_{\pm 0.35}$ | $72.60_{\pm 0.19}$ | $71.17_{\pm 0.02}$ | $70.99_{\pm 0.21}$ | $63.22_{\pm 0.72}$ | $73.52_{\pm 0.05}$ | $70.27_{\pm 1.19}$ |
| SFT | $70.90_{\pm 0.12}$ | $73.17_{\pm 0.21}$ | $70.89_{\pm 0.28}$ | $71.47_{\pm 0.25}$ | $65.75_{\pm 0.53}$ | $74.68_{\pm 0.23}$ | $71.09_{\pm 0.02}$ |
| Skywork-8B | $72.88_{\pm 0.19}$ | $74.56_{\pm 0.16}$ | $72.81_{\pm 0.89}$ | $72.40_{\pm 0.05}$ | $69.68_{\pm 0.56}$ | $76.00_{\pm 1.12}$ | $73.42_{\pm 0.09}$ |
| $\hookrightarrow$RankGRPO | $73.64_{\pm 0.30}$ | $75.25_{\pm 0.68}$ | $73.71_{\pm 0.46}$ | $73.77_{\pm 0.04}$ | $69.81_{\pm 0.40}$ | $\underline{77.22}_{\pm 0.25}$ | $73.57_{\pm 0.07}$ |
| URM-8B | $73.12_{\pm 0.28}$ | $75.14_{\pm 0.21}$ | $73.65_{\pm 0.84}$ | $73.47_{\pm 0.54}$ | $69.06_{\pm 0.68}$ | $76.16_{\pm 0.25}$ | $72.25_{\pm 1.05}$ |
| $\hookrightarrow$RankGRPO | $\underline{74.71}_{\pm 0.32}$ | $\underline{76.46}_{\pm 0.35}$ | $\underline{75.47}_{\pm 0.33}$ | $\underline{75.30}_{\pm 0.51}$ | $\underline{70.86}_{\pm 0.47}$ | $77.08_{\pm 0.23}$ | $\underline{73.72}_{\pm 0.11}$ |
| RRM(ours) | $\mathbf{81.33}_{\pm 0.12}$ | $\mathbf{83.27}_{\pm 0.35}$ | $\mathbf{82.92}_{\pm 0.18}$ | $\mathbf{81.68}_{\pm 0.23}$ | $\mathbf{75.90}_{\pm 0.60}$ | $\mathbf{84.16}_{\pm 0.09}$ | $\mathbf{80.96}_{\pm 0.63}$ |

(a) Math

(b) Writing

(c) Cross-Domain RM Comparison in Writing Benchmark

Figure 4: Reward-Guided Best-of-N Test-Time Scaling for Enhanced Inference Performance.

**Performance.** Table 2 presents the performance of RankGRPO on open-ended tasks. By converting absolute scalar scores from two distinct reward models into relative preferences through the *Ranking as Reward* approach, we achieve improvements across most domains. This highlights the effectiveness of relative ranking in enhancing both the stability and performance of GRPO. Moreover, when RankGRPO is paired with our fine-tuned RRM, it achieves the best results in the majority of domains, underscoring that evaluating the preference order of multiple responses provides a more robust learning signal than scoring individual responses. These findings reinforce the value of ranking-based approaches for improving performance in open-ended tasks.

## 5.3 EVALUATION OF RANKING REWARD MODEL

Figure 4 presents a comparative analysis between the SRM Skywork-Reward-V2-Llama-3.1-8B model (Liu et al., 2025a) and our proposed RRM framework under two experimental configurations.

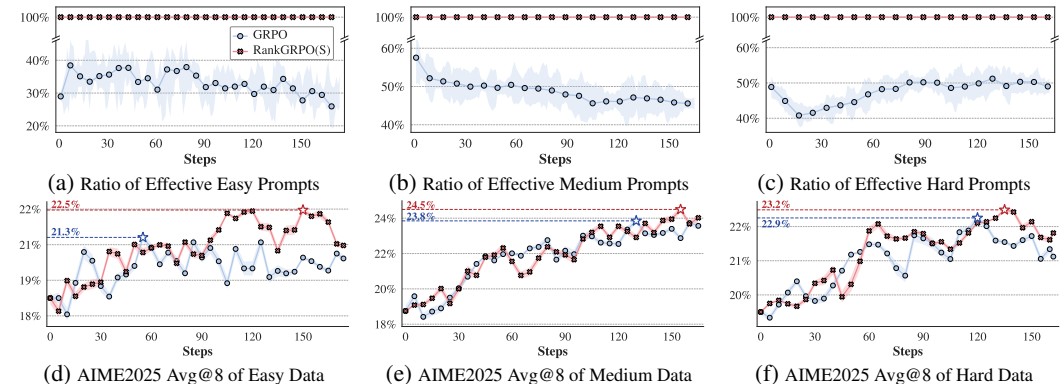

Figure 5: Comparison of GRPO and RankGRPO Across Different Data Difficulty Levels.

Table 3: Ablation study results on Responses Re-ranking. **Bold** marks the best result in each column.

| METHOD | MATHEMATICAL REASONING | | | | | | | | Avg | Avg Len. |
|---|---|---|---|---|---|---|---|---|---|---|
| | AIME24 | AIME25 | MATH500 | GSM8k | Olympiad | GaoKao | Minerva | AMC | | |
| RankGRPO (S) | 30.8 | 24.6 | 84.0 | 86.1 | 46.7 | **74.3** | 27.6 | **69.2** | 55.4 | 6365 |
| w/o Correctness | 28.4 | 23.2 | 83.9 | 86.2 | 45.1 | 73.9 | 27.7 | 66.7 | 54.4 | **5637** |
| w/o Length | **31.3** | **24.8** | **85.1** | **86.9** | **47.7** | 73.8 | **28.0** | 66.9 | **55.6** | 8452 |

Figure 1 demonstrates their performance on mathematical reasoning tasks using the DeepSeek-R1-Distill-LlaMA-8B model (Guo et al., 2025), where we sample $k$ responses per prompt and select via either SRM or RRM with Majority voting. The results show RRM achieves comparable or superior accuracy to SRM, with the performance gap widening as $k$ increases. Figure 4a and 4b illustrate RRM's superior performance in writing tasks evaluated on Qwen3-1.7B (Yang et al., 2025), where we sample eight responses per instance. The *Second* designation denotes the second-highest-scoring sample. RRM consistently selects higher-quality responses, explaining its significant improvement in compositional tasks. We attribute these improvements to RRM's focus on relative intra-group quality assessment rather than absolute scoring. Notably, RRM demonstrates strong generalization across domains despite limited training data.

## 5.4 METHOD ANALYSIS

We conduct a comprehensive analysis of our proposed method from several perspectives. More detailed results can be found in Appendix C.

### 5.4.1 IMPACT OF DATASET DIFFICULTY

The dataset difficulty directly affects the proportion of effective prompts during training. To examine this, we fine-tune a 1.5B model on three levels: (1) Easy: GSM8k (Cobbe et al., 2021), (2) Medium: SimpleRL (Zeng et al., 2025), and (3) Hard: Open-RS (Dang & Ngo, 2025).

Figure 5 presents the performance of GRPO and RankGRPO(S) across datasets of varying difficulty. Our method makes full use of the available data: on the easy dataset, RankGRPO(S) further improves model performance, whereas GRPO shows limited gains. On medium and hard datasets, both methods benefit from more informative prompts, and RankGRPO(S) consistently outperforms GRPO. The fraction of effective prompts under GRPO remains relatively low and exhibits difficulty dependent dynamics, while RankGRPO(S) maintains full utilization of groupwise information. Further analysis of these effects is provided in Appendix C.2.

## 5.5 ABLATION AND SENSITIVITY ANALYSIS

**Effect of Responses Re-ranking.** We analyze the contribution of different components in the re-ranking stage. Table 3 shows that removing correctness consistently degrades performance, confirming its necessity for reliable reasoning. Interestingly, relaxing the length constraint slightly im-

proves results on several datasets, suggesting that over-restricting output length may hinder problem-solving flexibility. Overall, correctness serves as the primary factor for stability, while length control requires a careful balance between conciseness and expressiveness.

**Effect of $\tau$ and Advantage Clipping.** We examine the sensitivity of model performance to different values of $\tau$. As shown in Figure 6, increasing $\tau$ weakens the influence of rule-based rewards, which in turn increases the likelihood of gradient directions misaligned with correctness, ultimately leading to performance degradation. Introducing clipping effectively mitigates this issue, resulting in more stable and robust performance improvements.

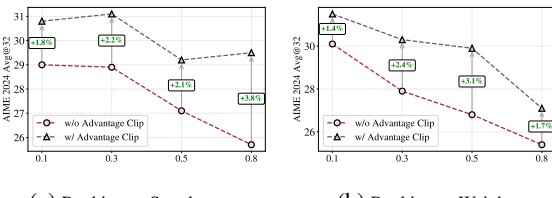

(a) Ranking as Supplement  (b) Ranking as Weight

Figure 6: Analysis of $\tau$ and Advantage Clipping.

## 6 CONCLUSION

We proposed RankGRPO, a reinforcement learning framework that utilizes intra-group preference rankings to overcome the challenges of sparse and unstable reward signals in LLM training. By integrating relative preferences through the RRM and three distinct strategies, RankGRPO mitigates the issues of vanishing gradients caused by identical rewards and reduces instability inherent in absolute numerical scoring. Experiments demonstrate that RankGRPO leads to improvements across multiple tasks, achieving consistent gains in RLVR and open-ended writing. These results emphasize that preference-based reward signals are more effective than absolute scoring systems in enhancing model performance. Additionally, the RRM, serving as a listwise ranking model explicitly aligned with group-normalized optimization, is well-suited for GRPO, achieving results comparable to SRM with fine-tuning on a small dataset. Future work will explore more efficient ways to leverage relative ranking rewards and investigate the potential of a more universal RRM to achieve broader generalization across diverse domains.

## ETHICS STATEMENT

Our method is based on fine-tuning open-source Large Language Models (LLMs) for tasks involving mathematical reasoning, logical reasoning, and open-ended writing. These tasks do not involve ethical concerns directly related to moral or societal norms. We have taken care to ensure that our experiments and methodologies adhere to responsible research practices, and our work does not involve harm to individuals or communities. The datasets and models used are publicly available, and we emphasize the transparency and reproducibility of our research.

## REPRODUCIBILITY STATEMENT

We have made our work fully reproducible by providing the source code, which is based on open-source frameworks with minimal modifications. The code is designed to be easily understandable and user-friendly. All experimental setups, model implementations, and training scripts are included, and detailed instructions are provided to facilitate replication of the results.

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

## A    THE USE OF LARGE LANGUAGE MODELS

We employed Large Language Models (LLMs) and Multimodal Large Language Models (MLLMs) exclusively for linguistic refinement of the manuscript, minor stylistic adjustments, and visual assistance in figure composition. The LLM/MLLM systems were not utilized in any aspect of research conception, experimental design, implementation, data analysis, or results interpretation. All technical content, conceptual frameworks, and substantive contributions originate entirely from the authors.

## B    TRAINING DETAILS

### B.1    SETTINGS

We cap the generated output length at 8,192 tokens and form groups of size $G = 8$ per prompt. For the RRM, we set the sortable subset size to $n = 4$. Unless otherwise noted, hyperparameters are fixed as follows: $\lambda = 2048$, $\xi^+/\xi^- = \pm 10^{-3}$ (advantage clipping threshold), $\tau = 0.1$ , and sampling temperature $T = 1.0$ during data collection. Our method and all baselines are implemented on top of the VeRL (Sheng et al., 2025) framework.

For the reward model evaluation, we set $n = 4$. When $n = 2$, we repeat the process once for each sample. For $n = 8$ or $n = 16$, we first divide the samples into multiple groups of size 4, then select the best from each group. Afterward, we continue with the RRM for a second round of selection until the optimal answer is chosen.

### B.2    TRAINING DYNAMIC

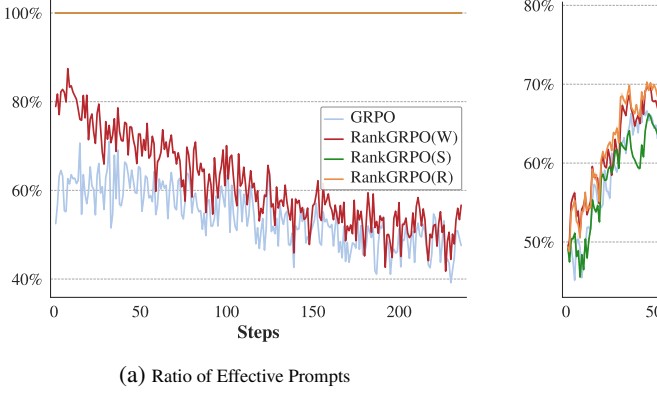

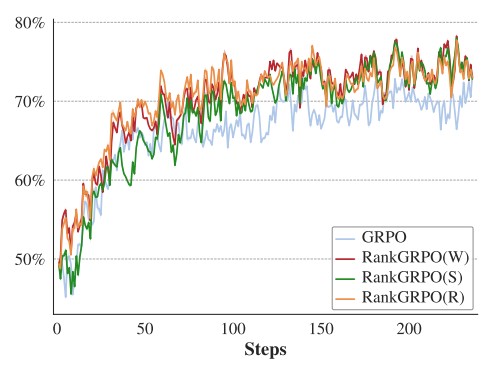

(a) Ratio of Effective Prompts

(b) Rule-base Rewards

Figure 7: Dynamic in key metrics during the training process of 8B Model.

We recorded the dynamic changes of two key metrics during the training process, as shown in Figure 7. The methods *Ranking as Supplement* and *Ranking as Reward* utilized all available data, with the proportion of effective data consistently remaining at 100%. In contrast, the *Ranking as Weight* method only used correctly grouped data, and as training progressed, the proportion of effective data gradually increased. All three methods, benefiting from additional knowledge, exhibited a slight advantage in reward scores compared to GRPO. This phenomenon highlights the effectiveness of the proposed relative ranking reward approach.

### B.3    EVALUATION DATASETS

We evaluate our models on seven mathematical reasoning benchmarks: Math500 (Hendrycks et al., 2021; Lightman et al., 2023a), AIME24 (Art of Problem Solving, 2024a), AIME25 (Art of Problem Solving, 2025), AMC (Art of Problem Solving, 2024b), Minerva Math (Lewkowycz et al., 2022), Gaokao (Zhang et al., 2023), and Olympiad Bench (He et al., 2024), which cover a broad range of mathematical difficulty and problem types. For logical reasoning, we select two representative

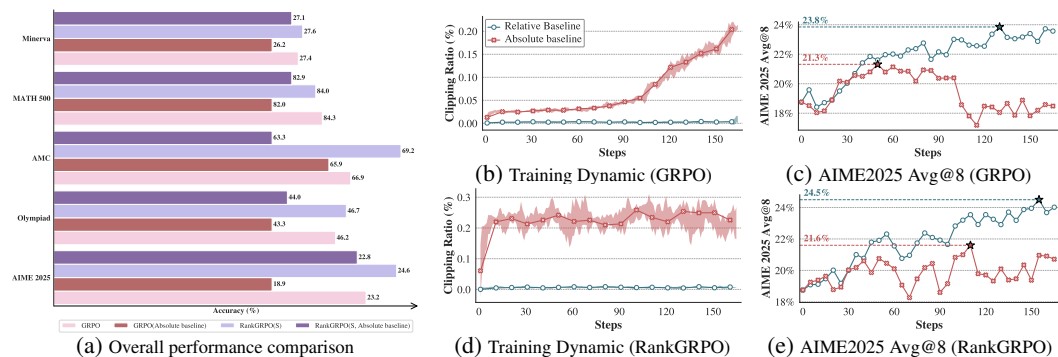

(a) Overall performance comparison     (b) Training Dynamic (GRPO)     (c) AIME2025 Avg@8 (GRPO)

(d) Training Dynamic (RankGRPO)     (e) AIME2025 Avg@8 (RankGRPO)

Figure 8: Effect of Absolute vs. Relative Baselines on GRPO and RankGRPO(S).

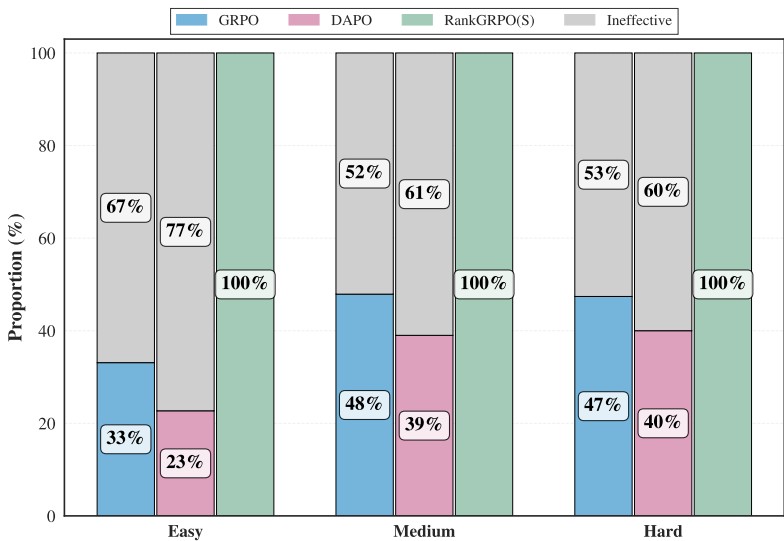

Figure 9: The proportion of effective data during the training phase for different methods.

benchmarks: Zebra Puzzle (Cheng et al., 2025), and Ordering Puzzle (Cheng et al., 2025). These datasets are widely recognized and present diverse challenges for evaluating both mathematical and general reasoning abilities.

## C  SUPPLEMENTARY RESULTS

### C.1  ABSOLUTE BASELINE.

In our experiments, we compared the impact of using relative baselines (intra-group mean) versus absolute correctness baselines (for GRPO, a baseline of 0.5; for RankGRPO, a baseline of 1) on performance and stability. The results, shown in Figure 8a, indicate that using the absolute correctness baseline leads to a significant drop in performance. Figure 8b and 8d further reveal the instability introduced by the absolute baseline, particularly from the perspective of truncation rates. Additionally, Figure 8c and 8e demonstrate a decline in accuracy during the later stages of training, highlighting the unsuitability of the absolute baseline for long-term training.

### C.2  IMPACT OF DATASET DIFFICULTY

We analyze how dataset difficulty influences data efficiency and performance in Section 5.4.1. Table 4 reports a detailed comparison of three methods across difficulty levels. Moderate difficulty yields the best gains, whereas overly easy or overly hard data diminishes further improvement. In

Table 4: Performance comparison across datasets of varying difficulty; **bold** indicates the best result.

| | Method | AIME24 | AIME25 | MATH500 | GSM8k | Olympiad | GaoKao | Minerva | AMC | Avg |
|---|---|---|---|---|---|---|---|---|---|---|
| Easy | GRPO | 29.4 | 23.0 | 83.1 | **87.0** | **45.1** | 72.5 | **28.4** | 64.3 | 54.1 |
| | DAPO | **30.9** | 23.0 | 83.1 | 86.0 | 43.9 | 73.4 | 26.5 | 64.1 | 53.9 |
| | RankGRPO(S) | 30.0 | **23.3** | **83.5** | 86.8 | 44.5 | **74.6** | 27.5 | **65.1** | **54.4** |
| Medium | GRPO | 29.7 | 23.8 | **84.3** | 85.6 | 46.2 | 73.9 | 27.4 | 66.9 | 54.7 |
| | DAPO | 30.4 | 23.3 | 83.2 | **86.1** | 43.7 | 74.1 | 26.8 | 65.3 | 54.1 |
| | RankGRPO(S) | **30.8** | **24.6** | 84.0 | **86.1** | **46.7** | **74.3** | **27.6** | **69.2** | **55.4** |
| Hard | GRPO | 27.9 | 23.6 | 82.9 | 86.0 | 43.4 | 74.2 | 26.3 | 65.4 | 53.7 |
| | DAPO | **29.0** | 22.5 | 83.5 | 85.8 | 44.2 | 72.9 | **27.4** | 63.8 | 53.6 |
| | RankGRPO(S) | 28.8 | **23.8** | **83.6** | **86.2** | **45.1** | **74.9** | 26.7 | **66.4** | **54.4** |

GRPO, extremes of difficulty tend to degenerate into *invalid prompts* that provide little learning signal and mainly act as a weak regularizer to prevent forgetting of trivial cases. In fact, on the easy subset many prompt groups are already close to unanimously correct at the beginning of training, so the GRPO effective prompt ratio starts at a relatively low level and quickly saturates. On the medium subset the GRPO effective prompt ratio decreases from about 60% at the beginning of training to about 40% near convergence, which is consistent with the global trend in Figure 1a. On the hard subset many prompt groups are initially unanimously incorrect and gradually become effective as the policy improves, which compensates for prompts that later turn unanimously correct and produces an almost flat curve. Across all difficulty levels, the absolute fraction of effective prompts under GRPO remains relatively low, indicating limited utilization of the available data. DAPO removes such invalid prompts altogether, which avoids noise but forfeits potential information contained therein. By contrast, RankGRPO leverages *all* samples by converting groupwise orderings into usable signal, thereby extracting additional knowledge even from otherwise low-value prompts. Figure 9 visualizes the fraction of effective data throughout RL training: RankGRPO maintains 100% effective utilization at all times, substantially exceeding the other methods and corroborating its advantages in both data efficiency and final performance.

## C.3 EFFECT OF GROUP SIZE ON METHOD PERFORMANCE

Table 5: Performance Comparison Across Different Group Sizes

| $G$ | Method | MATHEMATICAL REASONING | | | | | | | | |
|---|---|---|---|---|---|---|---|---|---|---|
| | | AIME24 | AIME25 | MATH500 | GSM8k | Olympiad | GaoKao | Minerva | AMC | Avg |
| 4 | GRPO | 28.8 | 22.5 | 83.2 | 84.7 | 46.1 | 73.0 | 27.8 | 66.7 | 54.1 |
| | RankGRPO(S) | 31.5 | 24.1 | 84.4 | 85.1 | 45.1 | 73.8 | 27.7 | 65.9 | 54.7 |
| 8 | GRPO | 29.7 | 23.8 | 84.3 | 85.6 | 46.2 | 73.9 | 27.4 | 66.9 | 54.7 |
| | RankGRPO(S) | 30.8 | 24.6 | 84.0 | 86.1 | 46.7 | 74.3 | 27.6 | 69.2 | 55.4 |
| 16 | GRPO | 29.1 | 23.0 | 85.7 | 87.3 | 47.6 | 76.2 | 28.1 | 68.0 | 55.6 |
| | RankGRPO(S) | 30.4 | 23.5 | 85.9 | 86.9 | 47.2 | 75.0 | 28.3 | 67.9 | 55.6 |

We conducted a comprehensive analysis of both methods' performance across varying group sizes $G$, as detailed in Table 5. The experimental results reveal that GRPO exhibits performance improvement with increasing $G$, primarily due to reduced occurrence of invalid groups at larger group sizes. In contrast, RankGRPO demonstrates consistent effectiveness even at smaller $G$ values through full utilization of all available training data. The experimental findings demonstrate that GRPO attains performance levels comparable to RankGRPO when operating with sufficiently large group sizes, as the increased sampling capacity enables more comprehensive data utilization.

## C.4 EFFECT OF $\lambda$ ON METHOD PERFORMANCE

We analyzed the impact of the parameter $\lambda$ in Equation (7) on model performance, with the results presented in Table 6. As the length constraint was relaxed, the model's performance steadily improved. When $\lambda$ was large, the length constraint became ineffective in most cases, and the performance closely resembled that of not using the length reordering. On the other hand, since the

**Question**

Let $\triangle ABC$ be a right triangle with $\angle A = 90°$ and $BC = 38$. There exist points $K$ and $L$ inside the triangle such that $AK = AL = BK = CL = KL = 14$. The area of the quadrilateral $BKLC$ can be expressed as $n\sqrt{3}$ for some positive integer $n$. Find $n$.

**RankGRPO[CoT Length: 6136 tokens]**

Given a right triangle $\triangle ABC$ with $\angle A = 90°$ and hypotenuse $BC = 38$. Points $K$ and $L$ inside the triangle satisfy $AK = AL = BK = CL = KL = 14$. We need to find the area of quadrilateral $BKLC$ expressed as $n\sqrt{3}$ and determine $n$.

1. **Identify the coordinates of points $K$ and $L$:**
   - Using the given conditions, we find the coordinates of $K$ and $L$ by solving the equations derived from the distances and the geometry of the triangle.
   - Points $K$ and $L$ are determined to be $K(2, 8\sqrt{3})$ and $L(13, 3\sqrt{3})$.
2. **Use the shoelace formula to find the area of quadrilateral $BKLC$:**\n - The coordinates of points $B(0, 16\sqrt{3})$, $K(2, 8\sqrt{3})$, $L(13, 3\sqrt{3})$, and $C(26, 0)$ are used.
   - Apply the shoelace formula to calculate the area:\n - Compute the sums Sum1 and Sum2 for the coordinates.
   - Calculate the absolute difference and divide by 2 to find the area.
3. **Calculate the area using the shoelace formula:**
   - Compute the sums for the shoelace formula:
   - Sum1 $= 422\sqrt{3}$
   - Sum2 $= 214\sqrt{3}$
   The area is $\frac{|422\sqrt{3} - 214\sqrt{3}|}{2} = 104\sqrt{3}$
   Thus, the value of $n$ is $\boxed{104}$

**GRPO[CoT Length: 7082 tokens]**

Given a right triangle $\triangle ABC$ with $\angle A = 90°$ and hypotenuse $BC = 38$. Points $K$ and $L$ inside the triangle satisfy $AK = AL = BK = CL = KL = 14$. We need to find the area of quadrilateral $BKLC$.

1. **Coordinates and Equations:**
   - Assign coordinates: $A(0, 0)$, $B(b, 0)$, $C(0, c)$ with $b^2 + c^2 = 38^2 = 1444$.
   - Points $K$ and $L$ have coordinates $(x_1, y_1)$ and $(x_2, y_2)$ respectively.
   - Using the given distances, we derive equations for $x_1, y_1, x_2, y_2$ and solve for $b$ and $c$.\n\n
2. **Solving for $b$ and $c$:**
   - Through geometric reasoning and solving the equations, we find $b = 26$ and $c = 16\sqrt{3}$.
3. **Coordinates of $K$ and $L$:**
   - $K$ is at $(13, 3\sqrt{3})$ and $L$ is at $(2, 8\sqrt{3})$.
4. **Area Calculation using Shoelace Formula:**
   - Vertices of quadrilateral $BKLC$ are $B(26, 0)$, $K(13, 3\sqrt{3})$, $L(2, 8\sqrt{3})$, and $C(0, 16\sqrt{3})$.
   - Applying the shoelace formula, the area is calculated as $104\sqrt{3}$.
   Thus, the area of quadrilateral $BKLC$ is $104\sqrt{3}$, so $n = \boxed{104}$

Figure 10: Performance on mathematical reasoning tasks, highlighting fewer reasoning tokens and clearer solution paths.

length constraint was applied only to correct responses, even with a small $\lambda$, there was no significant impact on the model's performance.

Table 6: Sensitivity Analysis of $\lambda$. $+\infty$ means RankGRPO(S) w/o Length Re-ranking

| $\lambda$ | Benchmark Scores | | | | | | | | Avg |
|---|---|---|---|---|---|---|---|---|---|
| | AIME24 | AIME25 | MATH500 | GSM8k | Olympiad | GaoKao | Minerva | AMC | |
| 512 | 29.2 | 25.2 | 84.1 | 86.7 | 45.6 | 75.5 | 26.0 | 65.7 | 54.7 |
| 1024 | 30.5 | 23.9 | 84.5 | 86.3 | 47.6 | 74.2 | 28.4 | 66.5 | 55.2 |
| 2048 | 30.8 | 23.6 | 84.0 | 86.1 | 46.7 | 74.3 | 27.6 | 69.2 | 55.3 |
| $+\infty$ | 31.3 | 24.8 | 85.1 | 86.9 | 47.7 | 73.8 | 28.0 | 66.9 | 55.6 |

## C.5 CASE STUDY

We present the performance on mathematical data in Figure 10. Since our method encourages exploring the optimal reasoning path while ensuring correctness, the number of reasoning tokens is relatively low, and the solution approach is clearer.

