# OpenReview forum: "Ranking is Reward: Intra-Group Preference Ranking for Group Relative Policy Optimization"
_ICLR.cc/2026/Conference — Submitted to ICLR 2026_

### Official Review · Reviewer_cfs1 · 2025-10-24

**Soundness:** 2
**Presentation:** 1
**Contribution:** 2
**Rating:** 2
**Confidence:** 3

**Summary:**

This work introduces a new variant of GRPO to be used for RL training of LLMs. The key proposal is that instead of simply using the reward, we also use the intra-group ranking as a way of augmenting the reward signal. Experimental validation shows a modest improvement on standard benchmarks.

**Strengths:**

I like the overall idea - relative ordering of answers can still provide a useful signal in GRPO-style algorithms. The paper is well-written in terms of formatting and overall flow.

**Weaknesses:**

1. There are no confidence intervals in any tables. This is a major issue, as the difference between 52.1 and 51.8 could be a meaningful (if small) improvement, or it could be completely random noise.
2. The procedure of training an RRM is not sufficiently explained, and seemingly only detailed in Figure 2?
3. The explanation of the proposed method is unclear. For example, Section 4.1 - what is “score”? Is it meant to be the reward? Advantage? Also $r_i$ is not defined, from the context I'm guessing it's the responses' ranking within the group, as evaluated by the RRM?
4. RRM - are we just distilling from the model’s token-based predictions?
5. What’s a “rule-based reward model”? Is it just a verifier? Or is it actually a learned reward model?
6. Section 4.3 - seems to imply that the final ranking is just correctness + length? I was hoping step 6 in Algorithm 1 would clarify this, but it remains vague
7. Section 4.2 - authors claim to propose a novel paradigm of RRM, but do not elaborate on how it works. A figure is not sufficient.
8. Experiments are only done on DeepSeek Distill models, which is a big limitation, and contamination is unknown.


Nitpicks:
1. Line 129/130 - very awkward phrasing, "direct parameter assignment"
2. Figure 1 is unclear. For example, looking at Fig. 1b, I’m thinking “What is it measuring? It it better high or low? Stable or unstable?”

**Questions:**

Questions/suggestions:
1. Improve the clarity of writing - there's a lot of under-described assumptions on how the RRM is trained or what the final rewards/advantages are.
2. Add confidence intervals. It seems like the only major difference in results between RankGRPO and baselines is on the "Logic Reasoning" benchmarks, but it's hard to tell right now - a discussion of this would be valuable.
3. What is the computational cost of this new method, compared to the baselines?

I'll increase my rating if the main points (clarity of writing, and confidence intervals) are sufficiently addressed

---

> ### Author Response · Authors · 2025-11-24
> **Weakness [W1 & W2 & W7 & W3]**
>
> > W1: There are no confidence intervals in any tables. This is a major issue, as the difference between 52.1 and 51.8 could be a meaningful (if small) improvement, or it could be completely random noise.
>
> ---
>
> We appreciate the reviewer’s emphasis on statistical rigor. While high computational costs often limit large-scale reinforcement learning studies to single-seed runs as seen in recent works like GRPO and DAPO we fully agree that validating result stability is essential. We **have conducted additional multi-seed runs** and evaluations for the primary experiments in Table 1 and 2. These tables now include **confidence intervals** to demonstrate the reliability of our method.
>
> The updated results remain consistent with our initial conclusions. In the mathematical and reasoning tasks shown in Table 1 RankGRPO yields **superior outcomes** by effectively learning from invalid samples. Additionally our ranking mechanism favors conciseness and results in a **significantly lower average token count** compared to other methods.
>
> Regarding open-ended tasks Table 2 presents the updated performance of RankGRPO. By converting absolute scalar scores from two distinct reward models into relative preferences via the **Ranking as Reward** approach we observe improvements across most domains. This highlights the effectiveness of relative ranking in enhancing the stability of GRPO. Furthermore RankGRPO achieves the **best results** in the majority of domains when paired with our fine-tuned RRM. This underscores that evaluating the **preference order** of multiple responses provides a more robust learning signal than scoring individual responses.
>
>
>
> ---
>
> W2 & W7
>
> > W2: The procedure of training an RRM is not sufficiently explained, and seemingly only detailed in Figure 2?
>
> > W7: Section 4.2 - authors claim to propose a novel paradigm of RRM, but do not elaborate on how it works. A figure is not sufficient.
>
>
> We appreciate the reviewer pointing out the lack of textual detail regarding the Ranking Reward Model. We acknowledge that relying primarily on Figure 2 was insufficient to explain the training procedure. In the revised manuscript we **have significantly revised Section 4.2** to formally define the mechanism and training objective of RRM.
>
> To clarify the working principle RRM differs from conventional reward models that output independent scalar scores. Instead RRM treats a group of $n$ responses as a **single input** and predicts a **permutation** over these candidates. Specifically we replace the language model head with a classification layer where each class corresponds to one of the **$n!$ possible orderings**. The model is trained using a **cross-entropy loss** on these permutations. This **listwise formulation** removes the need to calibrate scalar magnitudes and aligns explicitly with the group normalization structure of GRPO. We **have also refined** the phrasing "novel paradigm" to more accurately describe RRM as a **ranking-based reward model** designed to stabilize group-normalized policy optimization.
>
>
>
> > W3: The explanation of the proposed method is unclear. For example, Section 4.1 - what is “score”? Is it meant to be the reward? Advantage? Also $r_i$ is not defined, from the context I'm guessing it's the responses' ranking within the group, as evaluated by the RRM?
>
>
> We appreciate the reviewer pointing out the need for clearer definitions in Section 4.1. We acknowledge that the notation $r_i$ was ambiguous in the original manuscript as it referred to different rankings in different contexts. We have resolved this issue by distinguishing the notations strictly in the revised text. The term "score" is now defined as the **scalar reward** derived from rule-based verification. We introduced $r_i^{rrm}$ to denote the relative ranking evaluated by the RRM and reserved $r_i$ to represent the final relative ranking within the group after re-ranking.

---

> ### Author Response · Authors · 2025-11-24
> **Weaknesses W4 & W5 & W6 & W8**
>
> > W4: RRM - are we just distilling from the model’s token-based predictions?
>
> We apologize for the lack of clarity regarding the RRM architecture in the original manuscript. We would like to clarify that RRM is **not a distillation of token-based predictions**. We have revised the manuscript to make its role precise.  Instead, RRM is a sequence classification model that takes multiple complete responses as input and predicts a class that represents their joint ranking. In other words, it operates at the sequence level and outputs a relative ordering over responses, rather than imitating the token prediction behavior of another model.
>
> ---
>
> > W5: What’s a “rule-based reward model”? Is it just a verifier? Or is it actually a learned reward model?
>
> We appreciate the opportunity to clarify this terminology. You are correct that the "rule-based reward model" refers to **a deterministic verifier with no learnable parameters**. It functions by symbolically comparing the generated answer against the ground truth. In our work, we adopted the verification implementation from the [Skywork-OR1](https://github.com/SkyworkAI/Skywork-OR1/blob/main/verl/utils/reward_score/deepscaler_math/math_reward.py). To avoid future ambiguity, we have renamed this component to "Rule-based Verifier" throughout the revised manuscript.
>
> ---
>
> > W6: Section 4.3 - seems to imply that the final ranking is just correctness + length? I was hoping step 6 in Algorithm 1 would clarify this, but it remains vague
>
> We clarify that the final ranking is not a simple summation but a **hierarchical sorting strategy**, as illustrated in Figure 2a.
>
> Specifically, the sorting priority is **Correctness $\rightarrow$ Length $\rightarrow$ RRM**. This means the RRM determines the relative order only when responses are tied in both correctness and length bins. We have updated Section 4.3 and explicitly rewrote Step 6 in Algorithm 1 to describe this lexicographical sorting logic clearly, ensuring the specific role of RRM is unambiguous.
>
> ---
>
> > W8: Experiments are only done on DeepSeek Distill models, which is a big limitation, and contamination is unknown.
>
> To demonstrate the generalizability of our approach and ensure the results are not specific to the DeepSeek-Distill family, we conducted additional experiments using **Qwen3-1.7B** and **Qwen3-14B**. We compared RankGRPO(S) against the GRPO baseline on these distinct architectures.
>
> The results are presented in the table below.
>
> | Model | Method | AIME  | AMC | Zebra | Ordering | Avg Length (Tokens) |
> | :--- | :--- | :--- |  :--- | :--- | :--- | :--- |
> | Qwen3-1.7B | GRPO | 29.2 |  69.5 | 36.6 | 84.8 | 5736.7 |
> | Qwen3-1.7B | RankGRPO(S) | 32.1 |  71.6 | 42.1 | 87.5 | 5622.6 |
> | Qwen3-14B | GRPO | 45.0 |  82.8 | 46.4 | 89.0 | 7115.2 |
> | Qwen3-14B | RankGRPO(S) | 46.0 |  83.7 | 48.5 | 90.0 | 6321.7 |
>
> These findings demonstrate that RankGRPO consistently yields superior performance compared to GRPO across different model families and scales. Furthermore, RankGRPO maintains a lower average token count, confirming its efficiency. This evidence suggests that the improvements are driven by the proposed ranking mechanism rather than specific characteristics or potential data leakage associated with the DeepSeek models.

---

> ### Author Response · Authors · 2025-11-24
> **Nitpicks &  Questions/Suggestions**
>
> > Line 129/130 - very awkward phrasing, "direct parameter assignment"
>
> We appreciate the reviewer for pointing out the awkward phrasing. We have revised the sentence in the manuscript to ensure smoother expression and greater clarity.
>
>
> > Figure 1 is unclear. For example, looking at Fig. 1b, I’m thinking “What is it measuring? It it better high or low? Stable or unstable?”
>
> We apologize for the confusion regarding Figure 1. We have updated the figure captions and the corresponding text in the Introduction to clearly define the metrics and their implications.
>
> *   **Fig. 1a (Effective Data Ratio):** This tracks the proportion of rollout groups containing both correct and incorrect responses. A low ratio indicates that most groups have zero variance (all correct or all wrong), providing no gradients. The figure highlights that standard GRPO wastes significant rollout resources as this ratio drops during training.
> *   **Fig. 1b (Relative Range):** This measures the relative range of reward scores ($R$). To answer your specific questions: **excessively high values** are detrimental as they indicate outliers that skew advantage estimation, while **near-zero values** are also undesirable as they imply a lack of discrimination (no learning signal). Furthermore, the sharp fluctuations observed in the SRM curve represent **instability**, which hinders convergence. The figure demonstrates that our ranking approach avoids these extremes, offering a stable optimization landscape.
> *   **Fig. 1c (Visual Metaphor):** This summarizes the comparison. Standard GRPO often faces "calm seas" (zero variance/no learning), and SRMs resemble "stormy seas" (high instability). RankGRPO ensures "smooth sailing" by providing consistent and stable gradients.
>
> ---
>
> > Q1: Improve the clarity of writing - there's a lot of under-described assumptions on how the RRM is trained or what the final rewards/advantages are.
>
> We appreciate your constructive feedback regarding the clarity of our presentation. We have extensively revised the manuscript to explicitly detail the RRM training procedure. Regarding the computation of final values, we clarify that the advantages follow the standard GRPO formulation (as shown in Eq. 1). However, the final rewards $s_i^{rank}$ are computed based on the re-ranked order using the mapping function $f(\cdot)$ (as defined in Eqs. 3, 4, and 5). We have also updated Algorithm 1 to clearly illustrate this process.
>
> ---
>
> > Q2: Add confidence intervals. It seems like the only major difference in results between RankGRPO and baselines is on the "Logic Reasoning" benchmarks, but it's hard to tell right now - a discussion of this would be valuable.
>
> Thank you for this suggestion. As detailed in our response to **Weakness 1**, we have added confidence intervals to the primary experimental results. We would like to highlight that RankGRPO achieves consistent accuracy improvements on mathematical benchmarks. Furthermore, to better underscore our contribution to efficiency, we have added a new column for "Average Length" in the revised manuscript. This demonstrates that our method yields superior performance while significantly prioritizing concise reasoning chains.
>
> ---
>
> > Q3: What is the computational cost of this new method, compared to the baselines?
>
> We appreciate the reviewer raising this question regarding computational cost. The primary difference in resource usage among the methods stems from the specific strategies used for reward calculation. Our approach incurs a slightly higher cost than rule-based baselines due to the requirement for model inference. However, it significantly enhances data efficiency by utilizing all generated rollouts rather than discarding samples with zero variance. Furthermore, our method is more efficient than the Scalar Reward Model which evaluates each response individually. The RRM achieves this by processing $n$ samples simultaneously in a single inference pass. The table below estimates the resources required for training 100 steps on A100 GPUs.
>
> | Method | Effective Data Ratio | Training Resources (GPU Hours) |
> | :--- | :--- | :--- |
> | Rule-based | 48% | 72.8h |
> | SRM | 100% | 85.5h |
> | **RRM (Ours)** | **100%** | **79.1h** |
>
> Thank you again for your thoughtful questions and feedback throughout the review process. If you have any additional concerns or would like clarification on other aspects of the work, we would be very happy to address them.

---

### Official Review · Reviewer_y55s · 2025-10-27

**Soundness:** 2
**Presentation:** 2
**Contribution:** 2
**Rating:** 4
**Confidence:** 3

**Summary:**

This paper studies reinforcement learning with verifiable rewards in large language
models and proposes a new reward assignment method based on ranking. The authors observe
that a popular algorithm, GRPO, provides little to no learning signal when reward scores lack
variability within a batch during stochastic optimization. To address this, they propose ranking
the examples according to a ranking reward model and incorporating this ranking into the
advantage calculation of the GRPO loss. Experiments explore several variations of the core
idea across mathematical reasoning domains and more open-ended tasks such as creative and
technical writing. The authors report consistent performance gains over GRPO baselines.

**Strengths:**

The paper proposes a natural yet previously unexplored direction: ranking-based rewards in
RLVR. The authors introduce several variations of the core method, which they later
systematically ablate in the experiments, and analyze the impact of dataset difficulty on their
findings. The experiments cover multiple model configurations.

**Weaknesses:**

Unfortunately, the empirical results do not appear strong enough—or statistically significant—to
fully support the paper’s main claims. In particular, most experiments seem to be based on
single-seed runs. Moreover, the results reported in Tables 1 and 2 do not appear statistically
significant, as the performance differences between methods are often only a few percentage
points. It would strengthen the work if the authors demonstrated consistent improvements over
the baselines across multiple random seeds.

**Questions:**

1. It is unclear what Fig. 1(a) shows. How exactly do you define what an effective sample is?
2. Suggestion: You should mention that you provide rankings from a ranking model in line 155. (I know that you explain this later in Section 4.2, but it should be mentioned already in line 155; otherwise, it is confusing.)
3. “As illustrated in Figure 2, RRM ranks nn input responses, where each class corresponds to a permutation order.” Does this mean that the output dimension of the ranking model grows exponentially with the group size of GRPO? This sounds very inefficient.
4. Line 248 appears to need some rewriting—it is currently unclear.
5. Line 262: $r_{\phi}$ and $r_{\psi}$ should be capitalized.
6. Fig. 3a: it is unclear what resources the models used (e.g., how much data, compute, etc.).

---

> ### Author Response · Authors · 2025-11-24
> **Weakness & Question 1-2**
>
> > Unfortunately, the empirical results do not appear strong enough—or statistically significant—to fully support the paper’s main claims. In particular, most experiments seem to be based on single-seed runs. Moreover, the results reported in Tables 1 and 2 do not appear statistically significant, as the performance differences between methods are often only a few percentage points. It would strengthen the work if the authors demonstrated consistent improvements over the baselines across multiple random seeds.
>
>
> We appreciate the reviewer’s emphasis on statistical rigor. While high computational costs often limit large-scale reinforcement learning studies to single-seed runs as seen in recent works like GRPO and DAPO we fully agree that validating result stability is essential. We **have conducted additional multi-seed runs** and evaluations for the primary experiments in Table 1 and 2. These tables now include **confidence intervals** to demonstrate the reliability of our method.
>
> The updated results remain consistent with our initial conclusions. In the mathematical and reasoning tasks shown in Table 1 RankGRPO yields **superior outcomes** by effectively learning from invalid samples. Additionally our ranking mechanism favors conciseness and results in a **significantly lower average token count** compared to other methods.
>
> Regarding open-ended tasks Table 2 presents the updated performance of RankGRPO. By converting absolute scalar scores from two distinct reward models into relative preferences via the **Ranking as Reward** approach we observe improvements across most domains. This highlights the effectiveness of relative ranking in enhancing the stability of GRPO. Furthermore RankGRPO achieves the **best results** in the majority of domains when paired with our fine-tuned RRM. This underscores that evaluating the **preference order** of multiple responses provides a more robust learning signal than scoring individual responses.
>
>
> ---
>
>
> > Q1: It is unclear what Fig. 1(a) shows. How exactly do you define what an effective sample is?
>
>
>
> We appreciate the reviewer regarding the definition of an effective sample. We apologize for the lack of clarity in the original Figure 1(a). We have revised both the figure and its caption to explicitly define this term.
>
> In our context an **effective sample** refers to a data group that generates a **non-zero policy gradient** under the GRPO algorithm. GRPO estimates the advantage $A_i$ by normalizing the reward $r_i$ relative to the group mean $\bar{r}$ and standard deviation $\sigma_r$ according to the formula
> $$
> A_i = \frac{r_i - \bar{r}}{\sigma_r}
> $$
> When all rollouts in a group receive **identical rewards** the standard deviation $\sigma_r$ becomes zero. Consequently the calculated advantage $A_i$ vanishes for every sample in that group. This results in **no effective gradient signal** for the policy update. Therefore we consider only groups with **non-zero variance** where rollouts receive differentiating scores as effective samples. We have updated the Intruction to reflect this definition clearly.
>
>
> ---
>
>
>
>
> > Q2: Suggestion: You should mention that you provide rankings from a ranking model in line 155. (I know that you explain this later in Section 4.2, but it should be mentioned already in line 155; otherwise, it is confusing.)
>
> Thank you for this helpful suggestion. We have revised the text in line 155 to explicitly mention that the rankings are derived from a ranking model. This modification ensures the context is clear to the reader before the detailed explanation in Section 4.2.

---

> ### Author Response · Authors · 2025-11-24
> **Questions 3-6**
>
> > Q3: “As illustrated in Figure 2, RRM ranks nn input responses, where each class corresponds to a permutation order.” Does this mean that the output dimension of the ranking model grows exponentially with the group size of GRPO? This sounds very inefficient.
>
>
> We acknowledge that the output dimension theoretically grows factorially with the group size. In practice we address this by **fixing $n=4$** and splitting rollouts into subgroups to create a manageable **24-class task**. Moreover increasing $n$ requires concatenating more responses into a single input which significantly challenges the **LLM's context processing capabilities** due to excessive sequence lengths.
>
> To validate our choice we compared RankGRPO(S) across different group sizes. As shown in Table A while performance is comparable $n=4$ provides **finer-grained ranking signals** and **higher training throughput** than smaller groups while avoiding the computational and context bottlenecks of larger ones. Furthermore we compared the resource consumption of different reward models during RL training. Table B estimates the resources required for training 100 steps on A100 GPUs. It demonstrates that our RRM achieves a **100% effective data ratio** with **fewer GPU hours** than the Scalar Reward Model. Future work will explore generative reward models to overcome the factorial bottleneck.
>
> **Table A: Ablation study on group size $n$**
>
> | Datasets | $n=2$ | $n=3$ | $n=4$ |
> | :--- | :--- | :--- | :--- |
> | AIME25 | 24.0 | 24.0 | 24.6 |
> | MATH500 | 85.1 | 84.8 | 84.0 |
> | Olympiad | 46.6 | 47.0 | 46.7 |
> | Minerva | 27.3 | 27.7 | 27.6 |
> | Avg | 45.7 | 45.8 | 45.7 |
>
> **Table B: Resource comparison**
>
> | Method | Effective Data Ratio | Training Resources (GPU Hours) |
> | :--- | :--- | :--- |
> | Rule-based | 48% | 72.8h |
> | SRM | 100% | 85.5h |
> | **RRM (Ours)** | **100%** | **79.1h** |
>
>
>
> > Q4: Line 248 appears to need some rewriting—it is currently unclear.
>
> We appreciate you pointing out the lack of clarity in this sentence. We have rewritten line 248 to improve its readability and ensure the intended meaning is conveyed accurately.
>
> > Q5: Line 262: $r_\phi$ and $r_\psi$ should be capitalized.
>
> Thank you for your careful reading. We have corrected the capitalization error in line 262 as requested.
>
> > Q6: Fig. 3a: it is unclear what resources the models used (e.g., how much data, compute, etc.).
>
> We appreciate your suggestion to clarify the experimental setup. To ensure transparency we **have conducted a comprehensive survey** of the training resources and data sizes for the baselines shown in Figure 3a. The detailed comparison is presented in Table C below.
>
> These statistics highlight the **significant efficiency advantage** of RankGRPO. Compared to the resource-intensive baseline STILL-3 our method achieves **higher average performance** while utilizing only roughly **53% of the training data** and **20% of the computational time**. Regarding the Open-RS series our approach achieves a **substantial performance improvement** of nearly **3 percentage points** despite the difference in hardware configurations.
>
> We **have revised** the relevant section in the manuscript to include these details and to provide a more accurate discussion on the trade-off between resource consumption and model performance.
>
> **Table C: Resource and Performance Comparison**
>
> | Model | Data Size | Hardware | Training Time | Performance (Avg) |
> | :--- | :--- | :--- | :--- | :--- |
> | Open-RS1 | 7k | 4 $\times$ A40 (48G) | 24h | 53.8 |
> | Open-RS3 | 7k | 4 $\times$ A40 (48G) | 24h | 52.6 |
> | STILL-3 | 30k | 8 $\times$ A100 (80G) | 150h | 55.4 |
> | RankGRPO(S)  | 16k| 8 $\times$ A100 (80G)| 30h | 55.6 |
>
>
> Thank you again for your thoughtful questions and feedback throughout the review process. If you have any additional concerns or would like clarification on other aspects of the work, we would be very happy to address them.

---

### Official Review · Reviewer_RsH8 · 2025-11-01

**Soundness:** 2
**Presentation:** 2
**Contribution:** 2
**Rating:** 4
**Confidence:** 4

**Summary:**

The paper introduces RankGRPO, an enhancement for Group Relative Policy Optimization (GRPO) algorithm. The methodology is designed to tackle two primary issues in RLHF: 1) the "sparse reward" problem in Reinforcement Learning with Verifiable Rewards (RLVR), where groups of samples with identical scores (e.g., all correct) produce zero variance and thus no learning signal , and 2) the "range instability" of traditional scalar reward models (SRMs). The proposed solution involves a Ranking Reward Model (RRM), which outputs a relative preference ranking for multiple responses , and three strategies (Ranking as Weight, Supplement, or Reward) to integrate this rank information into the GRPO advantage calculation.

**Strengths:**

1. Proposes a Solution to the "Sparse Reward" Problem in GRPO
2. Addresses Reward Signal Instability
3. Demonstrates Improved Performance Across Diverse Tasks

**Weaknesses:**

1. The paper claims that as GRPO training progresses, more sample groups become "unanimously correct," which causes zero variance and reduces the number of “effective training samples. Figure 1a visualizes this by showing the ratio of effective prompts for GRPO dropping to around 40%.. However, this key motivational claim is contradicted by the paper’s own later analyses in Figures 5. In Figures 5, they showed that the effective prompt ratio for GRPO on easy, medium, and hard datasets actually remains stable (rather than diminishing).
2. The RankGRPO framework is not a straightforward substitution of scores with ranks but a complex, multi-stage, heuristic-driven pipeline that covers the true source of its gains. Before any "Ranking as X" strategy is applied, the paper adds a "Re-ranking" phase that sorts responses by correctness, length, and an RRM-based quality rank. The justification for the length-based heuristic is weak, as the ablation study in Table 3 shows that removing it actually improves results across multiple benchmarks. Finally, the paper presents three non-equivalent ranking strategies (W, S, R), with different ones performing best for different tasks, which lack a unified and principled approach.
3. The paper introduces the RRM as a "novel paradigm" that outputs relative preference rankings, but this is essentially what standard RLHF preference models have done for years using pairwise or k-wise comparisons. While the authors frame RRM as an alternative to scalar reward models (SRMs), the more appropriate comparison is to modern preference-based models such as those used in DPO. Ultimately, the paper does not explain how RRM is fundamentally different from a conventional preference model trained to rank multiple outputs.

**Questions:**

1. The RRM training process (Section 4.2) seems to be unclear. It states responses are scored via a "rule-based reward model" and an "LLM to rank their relative quality". How are these two signals combined to create the training data? Is the LLM-as-judge only used when the rule-based scores are tied?
2. Figure 1b shows that "SRM w/ Ranking" is stable, unlike the "SRM" line. This implies a simple solution exists: just convert an existing SRM's scores to ranks. Why is a new RRM needed if a standard SRM's ranks are already stable? Can you compare this "SRM-Rank" baseline directly against the RRM?
3. A key claim is that RankGRPO learns from "all-correct" groups. In such a group, what is the RRM actually ranking?

---

> ### Author Response · Authors · 2025-11-24
> **Weakness 1**
>
> > W1: The paper claims that as GRPO training progresses, more sample groups become "unanimously correct," which causes zero variance and reduces the number of “effective training samples. Figure 1a visualizes this by showing the ratio of effective prompts for GRPO dropping to around 40%.. However, this key motivational claim is contradicted by the paper’s own later analyses in Figures 5. In Figures 5, they showed that the effective prompt ratio for GRPO on easy, medium, and hard datasets actually remains stable (rather than diminishing).
>
>
> Thank you for pointing out the inconsistency between Figure 1a and Figure 5. The apparent mismatch primarily comes from the y axis range in the original version of Figure 5, which makes the change on the medium subset visually subtle rather than indicating a contradiction in our conclusions.
>
> Our intention is to show that GRPO suffers from low effective sample utilization once many prompt groups become unanimously correct or unanimously incorrect. On the medium difficulty subset in Figure 5, the effective prompt ratio decreases from about 60 percent at the beginning of training to about 40 percent near convergence. This is consistent with the global behavior in Figure 1a and with Figure 7a, and it shows that a substantial portion of potentially informative prompts no longer contributes useful gradients.
>
> For the easy and hard subsets, the curves look flatter because of different dynamics. On the easy subset, many prompt groups start very close to unanimously correct, so the effective ratio is already low and quickly saturates. On the hard subset, many prompt groups are initially unanimously wrong and gradually become effective as the policy improves, which partially compensates for those that later turn unanimously correct. This compensation produces a visually stable curve.
>
> The key point is that **the absolute effective prompt ratio remains relatively low across difficulty levels**, especially on easy and medium prompts. The exact trend over training is secondary. What matters is that GRPO spends much of training in a regime where **only a limited portion of data is effective**, indicating poor utilization of available samples. In the revised manuscript, we have adjusted the y axis range in Figure 5 and added explicit clarifications in the Introduction and Section 5.4.1 so that this interpretation is clear and consistent across Figures 1, 5, and 7.

---

> ### Author Response · Authors · 2025-11-24
> **Weakness 2**
>
> > W2: The RankGRPO framework is not a straightforward substitution of scores with ranks but a complex, multi-stage, heuristic-driven pipeline that covers the true source of its gains. Before any "Ranking as X" strategy is applied, the paper adds a "Re-ranking" phase that sorts responses by correctness, length, and an RRM-based quality rank. The justification for the length-based heuristic is weak, as the ablation study in Table 3 shows that removing it actually improves results across multiple benchmarks. Finally, the paper presents three non-equivalent ranking strategies (W, S, R), with different ones performing best for different tasks, which lack a unified and principled approach.
>
>
>
> The core contribution of RankGRPO is **constructing advantages from groupwise relative rankings** rather than from raw scalar scores. The re-ranking phase acts as a **modular preference aggregation step** that integrates signals such as correctness and quality.
>
> Regarding the length-based heuristic, it encodes a simple preference where **a shorter chain of thought is prioritized among equally correct responses**. We have extended the ablation study in the revised manuscript to report **average output lengths**. The results show that RankGRPO with length-based re-ranking produces significantly shorter sequences while maintaining accuracy. This demonstrates that although removing the heuristic might slightly improve accuracy on some benchmarks, keeping it provides a clear efficiency gain.
>
> Furthermore, we wish to emphasize that RankGRPO is designed as a **general framework** rather than a single fixed algorithm. The three variants differ in how strongly they incorporate ranking information.
>
> *   **Ranking as Weight** uses ranking only to redistribute advantages among samples that already receive positive rule-based rewards.
> *   **Ranking as Supplement** combines rule rewards with ranking-based adjustments where rule rewards remain the primary signal.
> *   **Ranking as Reward** relies solely on the relative ranking inside each group rather than numerical values.
>
> We have added a discussion to the revised paper emphasizing that incorporating ranking into group-based optimization provides three key benefits.
>
> 1.  **Mitigates sparse-reward inefficiency**
>     Every sample in a group receives a meaningful training signal through its relative position.
> 2.  **Reduces reliance on unstable numerical scales**
>     Optimization depends on preference order rather than raw score magnitudes.
> 3.  **Enables principled multi-objective optimization**
>     RankGRPO naturally converts heterogeneous signals into a single unified ranking. This allows the policy to optimize multiple objectives such as correctness and efficiency in a controlled manner.

---

> ### Author Response · Authors · 2025-11-24
> **Weakness 3**
>
> > W3: The paper introduces the RRM as a "novel paradigm" that outputs relative preference rankings, but this is essentially what standard RLHF preference models have done for years using pairwise or k-wise comparisons. While the authors frame RRM as an alternative to scalar reward models (SRMs), the more appropriate comparison is to modern preference-based models such as those used in DPO. Ultimately, the paper does not explain how RRM is fundamentally different from a conventional preference model trained to rank multiple outputs.
>
>
> Thank you for raising this point about the positioning of RRM. We agree that RRM is more closely related to modern preference-based models than to traditional scalar reward models and we **have clarified** this in the revised manuscript. Our intention is not to claim a completely new family of models but to propose a **ranking-based reward model** that is tailored to the **group-normalized objective of GRPO**.
>
> Conventional RLHF reward models are typically trained from pairwise or k-wise preferences but at inference time they output **independent scalar scores** for each response. These scores may preserve the correct order yet their absolute magnitudes and variance are difficult to control. We **have emphasized** in the revised text that when such scalars are plugged into GRPO the group mean normalization becomes **sensitive to changes in score scale** and leads to **unstable advantages** even if the underlying preferences remain the same. In contrast, RRM does not output unconstrained scalars. It treats the entire group of $n$ responses as a **single input** and predicts a **permutation** over these $n$ candidates. Concretely, we replace the language model head with a classification layer whose classes correspond to the possible orderings and we train it with a cross-entropy loss on these permutations. This **listwise formulation** removes the need to calibrate scalar magnitudes and matches the relative nature of GRPO’s normalization.
>
> Regarding the relation to DPO-style methods, DPO uses pairwise preferences to define a loss directly on the policy probabilities and does not learn a separate reward model that can be applied to arbitrary groups of responses. RRM serves a different purpose. It is a **stand-alone ranking model** that can take any set of sampled responses for a prompt and output a consistent total order over them. In our training pipeline, we use rule-based correctness and an SRM only to construct supervision signals for RRM and then discard their raw scores. Once trained, RRM can be reused across policies and directly supplies the group-level ranking required by RankGRPO. In this sense, RRM is closer to a **listwise preference model** whose objective and output space are **explicitly aligned with the group structure of GRPO** rather than a conventional scalar reward model or a DPO-style implicit preference model.
>
> We **have softened** the wording around "novel paradigm" and **described** RRM as a ranking-based reward model designed for group-normalized policy optimization. We hope this makes clear that our contribution lies in identifying the mismatch between scalar rewards and GRPO and in showing that a listwise ranking model that outputs permutations rather than scalar scores leads to more stable and effective use of preference information within RankGRPO.

---

> ### Author Response · Authors · 2025-11-24
> **Questions**
>
> > Q1: The RRM training process (Section 4.2) seems to be unclear. It states responses are scored via a "rule-based reward model" and an "LLM to rank their relative quality". How are these two signals combined to create the training data? Is the LLM-as-judge only used when the rule-based scores are tied?
>
> Thank you for pointing out that the description of the RRM training process was ambiguous. In the original version, Figure 2 only showed an “LLM to rank quality”, which could be read as if we always used an LLM judge. In the revised version we clarify that the supervision for RRM is derived mainly from a scalar reward model, and that the LLM judge is only a fallback for rare inconsistent groups.
>
> For tasks with verifiable correctness such as math and logic, we first use a rule based verifier to assign a binary correctness label. We then apply an SRM to refine the order inside the correct group and inside the incorrect group, and we always enforce that every correct response must be ranked above every incorrect response. The LLM as judge is used only when the SRM ranking conflicts with correctness in a given group, for example when an incorrect response receives a higher SRM score than a correct one. In that case we discard the SRM scores for that group and ask the LLM judge to provide a local ranking that respects the correctness constraint. Therefore the LLM judge is not triggered by ties in the rule based scores, but specifically by conflicts between SRM and correctness.
>
> For open ended tasks where no reliable rule based verifier is available, we simply sort responses by SRM scores and do not use an LLM judge. We have updated Section 4.2 and Figure 2 to reflect this pipeline accurately.
>
> ---
>
> > Q2: Figure 1b shows that "SRM w/ Ranking" is stable, unlike the "SRM" line. This implies a simple solution exists: just convert an existing SRM's scores to ranks. Why is a new RRM needed if a standard SRM's ranks are already stable? Can you compare this "SRM-Rank" baseline directly against the RRM?
>
> Thank you for the insightful question regarding the “SRM w/ Ranking” curve in Figure 1b. We agree that converting SRM scores to within-group ranks already stabilizes GRPO and this is why we **have utilized** an SRM-Rank baseline in our analysis. The critical difference lies in the **ranking mechanism itself**.
>
> A standard SRM operates in a **pointwise fashion** where it scores each response independently. The groupwise order is obtained only by sorting these separate scores. In contrast, RRM is explicitly **listwise**. It takes all $n$ responses for a prompt as a **joint input** and predicts a permutation over these candidates in one forward pass. This enables the model to **directly compare multiple responses** during the inference process rather than inferring the order indirectly.
>
> We evaluate this SRM-Rank baseline in Figures 3b, 3c and 3d. The results demonstrate that **RRM performs consistently slightly better than SRM-Rank**. More importantly, **both ranking-based methods significantly outperform the standard SRM**. This confirms that the ranking formulation itself is the primary driver of training stability while the listwise architecture of RRM provides further refinement over the pointwise approach.
>
> ---
>
> > Q3: A key claim is that RankGRPO learns from "all-correct" groups. In such a group, what is the RRM actually ranking?
>
> In our terminology an “all-correct” group means that the rule-based verifier marks all responses as correct. However these responses still differ along several quality dimensions.
>
> The RRM is trained to rank based on finer-grained preferences. To construct supervision data for the RRM we employ a hybrid pipeline. We primarily rely on an SRM which is trained on large-scale preference datasets and effectively captures general human preferences. The LLM judge serves only as a fallback mechanism. It is invoked specifically when the SRM score contradicts the rule-based correctness constraint. In such cases we explicitly instruct the judge to rank solutions based on **logic**, **correctness of details** and **fluency**.
>
> Consequently the RRM provides a **non-trivial ranking** inside all-correct groups. This allows the algorithm to update the policy toward shorter and better-structured solutions instead of treating all correct responses as equally good. To validate this we conducted an ablation study comparing RankGRPO with RRM against a variant that uses a **random order** within all-correct groups. The results in the table below demonstrate that the random baseline underperforms. This confirms that the RRM rankings contain **meaningful learning signals** beyond binary correctness.
>
> ||AIME24|AIME25|Minerva|AMC|Avg|
> |-|-|-|-|-|-|
> |GRPO|29.7|23.8|27.4|66.9|37.0|
> |Random Rank w/ correctness rerank|29.9|24.1|27.2|68.3|37.4|
> |RRM|30.8|24.6|27.6|69.2|38.1|
>
> Thank you again for your thoughtful questions and feedback. If you have any further concerns or need clarification, we’d be happy to address them.

---

> ### Comment · Reviewer_RsH8 · 2025-11-28
>
> Dear AC and authors,
>
> Thank you for your efforts. I am unable to edit my score, possibly due to the ICLR and Open review system error. I believe that a score of **6: marginally above the acceptance threshold** is appropriate for this work, given that the authors have conducted extensive experiments and analyses.
>
> Thank you,
>
> Reviewer RsH8

---

### Official Review · Reviewer_isFw · 2025-11-03

**Soundness:** 3
**Presentation:** 3
**Contribution:** 3
**Rating:** 6
**Confidence:** 4

**Summary:**

The authors introduce RankGRPO, a framework in which a relative reward model (RRM) ranks responses in a group from which the rank is incorporated into the final reward used in advantage estimation within GRPO. In cases where only a sparse binary reward is available, the use of the RRM can provide more granular signal on the "better" responses. The RRM is trained on a dataset of rankings synthetically generated from an LLM.

**Strengths:**

- The experimental results are extensive and thorough, including ablations
- The paper is well-written and the figures are instructive

**Weaknesses:**

- There is a significant overhead in having to train the RRM - especially if it needs to be separately trained for each and every RL environment which would limit it's practical use (though whether this is true or not wasn't clear to me)
- The exact training procedure (e.g., loss function) for the RRM was unclear

**Questions:**

- Please address the weaknesses
- Does RankGRPO not apply in reward settings which are not correctness-based but are continuous (i.e., non-binary)?
- 3 variants of RankGRPO are provided (weight, supplement, reward) but it is not clear to me if one is universally better than the others? If not, are there certain scenarios where one variant is better than the other?

---

> ### Author Response · Authors · 2025-11-24
> **Weaknesses & Question 1**
>
> > W1: There is a significant overhead in having to train the RRM - especially if it needs to be separately trained for each and every RL environment which would limit it's practical use (though whether this is true or not wasn't clear to me)
>
>
> We thank the reviewer for raising this concern about the training overhead and practicality of the Ranking Reward Model RRM.
>
> First, **the computational cost of training RRM is comparable to standard RLHF reward models**. RRM is obtained by fine tuning the same LLM backbone, with a classification head trained using a standard cross entropy objective. In our setting it is in fact cheaper in practice because RRM uses **much less preference data**. While state of the art reward models such as Skywork Reward V2 [1] are trained on about 26 million preference pairs, we fine tune RRM on 18k group wise preference examples for mathematical and logical reasoning and 10k examples for open ended writing. This one time fine tuning cost is small relative to the cost of RL training.
>
> Second, in this work we train a small number of task specific RRMs rather than one model per benchmark. We use one RRM for the mathematical and logical reasoning domain and one RRM for open ended writing. The RRM for mathematical and logical reasoning is trained only on a subset of the [Open-R1-Math-220k dataset](https://huggingface.co/datasets/open-r1/OpenR1-Math-220k), which contains mathematical problems but no explicit logical reasoning tasks, yet RankGRPO with this RRM still outperforms other methods on logical reasoning benchmarks. This suggests that the ranking signal learned by RRM can transfer across related environments such as math and logic. We believe that extending this idea to more heterogeneous domains and investigating a more universal RRM is a promising direction for future work. We have clarified these points in the revised version.
>
>
> [1] Liu, Chris Yuhao, et al. Skywork-Reward-V2: Scaling Preference Data Curation via Human-AI Synergy. arXiv preprint arXiv:2507.01352, 2025, https://arxiv.org/abs/2507.01352.
>
> ---
>
> > W2: The exact training procedure (e.g., loss function) for the RRM was unclear.
>
> Conventional scalar reward models use a scalar head trained with a pairwise preference loss, typically a BCE-based formulation. In contrast, **RRM is a listwise sequence-classification model**. For each group of candidate responses, we construct a ranking label representing their preference order and fine-tune the classification head using a **standard cross-entropy loss** on this multi-class label.
>
> In the revised manuscript, we have updated the RRM subsection in the method section to explicitly state this training objective. We now clearly describe how the groupwise ranking labels are constructed from preference data and include the corresponding loss formulation so that the training procedure of RRM is fully specified.

---

> ### Author Response · Authors · 2025-11-24
> **Question 2 & 3**
>
> > Q2: Does RankGRPO not apply in reward settings which are not correctness-based but are continuous (i.e., non-binary)?
>
>
> Thank you for raising this important question.
>
> RankGRPO is **not limited to binary signals**. The algorithm only requires a **preference ordering** within each group, regardless of how this ordering is produced. In continuous-reward settings, although GRPO does not suffer from reward sparsity, it remains sensitive to numerical instability in scalar reward models. As shown in Figure 1(b), pairwise-trained scalar reward models can produce scores whose variance within a group is excessively high, occasionally exceeding eight times the group mean. Because GRPO computes advantages using the group mean, these extreme values directly distort advantage estimates and lead to unstable updates.
>
> To address this, RankGRPO can be applied by **converting continuous scalar rewards into induced rankings**. The continuous scores are first used to sort responses inside each group, and RankGRPO then operates on these relative ranks rather than raw scores. This removes the influence of poorly calibrated scales while still using all the information encoded in the continuous preferences.
>
> Table 2 reports the results on non binary reward tasks where the underlying reward models (such as Skywork Reward V2 and URM) output continuous scores. Methods that directly optimize on scalar rewards already improve over supervised baselines. After converting the same scores into relative rankings and applying RankGRPO, we observe consistent and stable gains over these scalar based methods. This demonstrates that RankGRPO remains effective in continuous reward settings and is not limited to correctness based binary rewards.
>
> ---
>
> > Q3: 3 variants of RankGRPO are provided (weight, supplement, reward) but it is not clear to me if one is universally better than the others? If not, are there certain scenarios where one variant is better than the other?
>
>
> Thank you for the question regarding the three variants of RankGRPO.
>
> The three variants differ in how strongly they incorporate ranking information:
>
> * **Ranking as Weight (W)**
>   Uses ranking only to redistribute advantages among samples that already receive positive rule-based rewards.
>
> * **Ranking as Supplement (S)**
>   Combines rule rewards with ranking-based adjustments. All samples in a group are separated by rank while retaining rule rewards as the primary signal.
>
> * **Ranking as Reward (R)**
>   Advantage computation relies solely on the relative ranking inside each group, rather than the numerical values of rule or scalar rewards.
>
> As shown in Table 1, for mathematical and logical reasoning tasks where rule rewards are reliable, **RankGRPO(S)** performs slightly better. This suggests that combining rule rewards with ranking information is particularly effective in such structured environments.
>
> Overall, RankGRPO is designed as a **general framework** rather than a single fixed algorithm. Incorporating ranking into group-based optimization provides three key benefits:
>
> 1. **Mitigates sparse-reward inefficiency**
>    Every sample in a group receives a meaningful training signal through its relative position.
>
> 2. **Reduces reliance on unstable numerical scales**
>    Optimization depends on preference order rather than raw score magnitudes.
>
> 3. **Enables principled multi-objective optimization with priority structure**
>    Traditional scalar reward modeling struggles to combine multiple objectives, such as correctness, reasoning efficiency, and chain-of-thought quality, when these objectives have **different priorities**. RankGRPO naturally converts multiple heterogeneous signals into a single unified ranking, allowing the RL policy to optimize them in a controlled, priority-aware manner.
>
> Thank you again for your thoughtful questions and feedback throughout the review process. If you have any additional concerns or would like clarification on other aspects of the work, we would be very happy to address them.

---

### Author Response · Authors · 2025-12-01
**Rebuttal Summary by the Authors**

Dear Area Chair and Reviewers,

As the active discussion phase was impacted by the recent system restrictions, we would like to provide a concise summary of our rebuttal to assist the Area Chair in their final assessment.

First, we sincerely thank all reviewers for their thoughtful comments and the time devoted to evaluating our work. In particular, we appreciate the careful scrutiny of our experimental rigor and baseline design. We have revised the manuscript and added new results that directly address the main concerns, which we believe strengthens both the reliability and the generality of the paper.

**Summary of Key Revisions and New Results**

* **Stronger Statistical Rigor (Re: y55s, cfs1).**
  We conducted multi-seed runs and evaluations for all primary experiments. The updated results show that RankGRPO’s improvements over GRPO are statistically significant and stable, alleviating concerns about variance and random noise.

* **Generalization Across Architectures (Re: cfs1).**
  In addition to the DeepSeek family, we added experiments on Qwen3 models. RankGRPO consistently outperforms GRPO across these architectures, indicating good generalization of the method.

* **Computational Efficiency (Re: isFw, y55s, cfs1).**
  We provided a resource and runtime analysis showing that RankGRPO is computationally efficient. The length-aware ranking also encourages more concise generations, which reduces inference latency in practice.

* **Methodological Clarifications and Exposition (Re: isFw, cfs1).**
  We improved the overall clarity of writing, updated Section 4.3 to describe the RRM training pipeline more concretely, and refined the description of Figure 1 as well as the axis range in Figure 5 for better interpretability. Most importantly, we clarified the design philosophy behind RankGRPO and the intended roles of its three variants, resolving earlier confusion about how the ranking signal is incorporated in each variant.

**Conclusion**

Reviewers have noted that the proposed ranking-based reward is a **“natural yet previously unexplored direction”** (y55s) supported by **“extensive and thorough”** experiments (isFw). With the additional multi-seed validation and cross-architecture results incorporated during the rebuttal, we believe the main concerns about statistical significance, baseline fairness, and model diversity have been fully addressed, an improvement also acknowledged by Reviewer RsH8 with a score increase to 6.


Thank you again for your time and consideration.

Best regards,

The Authors

---

### Meta-Review · Area_Chair_rwfV · 2025-12-28

**Summary:**

RankGRPO introduces intra-group preference ranking as a reward signal to address "sparse reward" and "range instability" issues in GRPO. The authors propose a listwise Ranking Reward Model (RRM) that predicts relative preference rankings for multiple responses simultaneously.

**Reviewer Concerns:**

Unaddressed ones:

Reviewer cfs1 pointed out that the "re-ranking" phase in the Supplement (S) and Reward (R) variants relies partly on a length-based heuristic to penalize verbosity. The response did not provide a theoretical justification for why this specific heuristic is optimal across all domains.

Reviewer isFw pointed out that the optimal strategy changes depending on the task. The authors did not provide a principled method for choosing a variant, leaving it as a manual trial-and-error process for users.

Reviewer y55s raised concerns about the mathematical complexity as group sizes ($G$) grow. Since the RRM compares all items in a group, the output space grows factorially; while the authors use sub-groups to mitigate this, the performance degradation at very high $G$ remains largely unexplored.

**Reviewer Scores:**

Reviewer isFw is likely to maintain 6, Reviewer RsH8 could increase to 6, Reviewer cfs1 will increase to 4 and Reviewer y55s could maintain 4 or increase to 6.

---

### Decision · Program_Chairs · 2026-01-26

Reject